# A time-resolved multi-omics atlas of transcriptional regulation in response to high-altitude hypoxia across whole-body tissues

Ze Yan[1,2,4], Ji Yang[1,2,4], Wen-Tian Wei[1,2], Ming-Liang Zhou[3], Dong-Xin Mo[1,2], Xing Wan[1,2], Rui Ma[1,2], Mei-Ming Wu[1,2], Jia-Hui Huang[1,2], Ya-Jing Liu[1,2], Feng-Hua Lv[1,2] & Meng-Hua Li[1,2] ✉

High-altitude hypoxia acclimatization requires whole-body physiological regulation in highland immigrants, but the underlying genetic mechanism has not been clarified. Here we use sheep as an animal model for low-to-high altitude translocation. We generate multi-omics data including whole-genome sequences, time-resolved bulk RNA-Seq, ATAC-Seq and single-cell RNA-Seq from multiple tissues as well as phenotypic data from 20 bio-indicators. We characterize transcriptional changes of all genes in each tissue, and examine multi-tissue temporal dynamics and transcriptional interactions among genes. Particularly, we identify critical functional genes regulating the short response to hypoxia in each tissue (e.g., *PARG* in the cerebellum and *HMOX1* in the colon). We further identify TAD-constrained *cis*-regulatory elements, which suppress the transcriptional activity of most genes under hypoxia. Phenotypic and transcriptional evidence indicate that antenatal hypoxia could improve hypoxia tolerance in offspring. Furthermore, we provide time-series expression data of candidate genes associated with human mountain sickness (e.g., *BMPR2*) and high-altitude adaptation (e.g., *HIF1A*). Our study provides valuable resources and insights for future hypoxia-related studies in mammals.

Hypoxia is a severe challenge to an organism's homeostatic equilibrium, affecting the physiological and pathological processes of organisms at high altitude[1,2]. The genetic mechanisms underlying long-term hypoxia adaptation have revealed positively selected genes and non-coding variants associated with cardiovascular, respiratory, and metabolic traits in highland human and other vertebrates[3–6], such as *EPAS1* and *EGLN* genes (e.g., *EGLN1*, *EGLN2* and *EGLN3*) in the hypoxia-inducible factor (HIF)-prolyl hydroxylase domain (PHD)-Von Hippel-Lindau (VHL) pathway[7]. However, in human and animals inhabiting lowlands, visible physiological adjustments in response to hypoxia (e.g., acute increase in ventilation) occur during short-term acclimatization after moving to a high altitude[8], whose mechanisms have not yet been elucidated. Both short-term acclimatization and long-term adaptation have a genetic background, thus they are genetically linked to each other[9]. For instance, differentially expressed genes (DEGs) identified during short-term acclimatization may have corresponding single-nucleotide polymorphisms (SNPs) or structural variants (SVs) between high-altitude and low-altitude populations during long-term adaptation[10]. Common signaling pathways may be enriched during short-term response and long-term adaptation to hypoxia[11]. In

[1]State Key Laboratory of Animal Biotech Breeding, China Agricultural University, Beijing 100193, China. [2]College of Animal Science and Technology, China Agricultural University, Beijing 100193, China. [3]Sichuan Academy of Grassland Science, Chengdu 611743, China. [4]These authors contributed equally: Ze Yan, Ji Yang. ✉e-mail: menghua.li@cau.edu.cn

the case of maladaptation, hypoxia can lead to high-altitude diseases, such as polycythemia, pulmonary hypertension, and heart failure in immigrants from lowland areas and chronic mountain sickness in high-altitude inhabitants[12–14]. Hypoxia also induces serious sickness in livestock transported to high-altitude areas, such as pulmonary hypertension in sheep[15] and brisket disease in cattle[16].

Sheep (*Ovis aries*) are an excellent model for studying hypoxia adaptation and acclimatization since they have adapted to a variety of environments (e.g., lowland and highland) around the world[17,18]. Compared with other large animals, such as non-human primates, dogs, pigs and horses, sheep are an applicable animal model for biomedical research in terms of cost-effectiveness and ethical isssues[19,20]. For example, sheep models have been widely used for studying human cardiopulmonary, respiratory, neurological, immunological, and reproductive diseases[21–25] and for fetal-neonatal development and pathologies[26]. Additionally, Dolly the sheep was the first animal to ever be successfully cloned from cultured somatic cells[27], making it possible to use the somatic cell nuclear transfer (SCNT) technique for both biomedical and agricultural applications of sheep[28].

In this study, we performed a low-to-high altitude translocation experiment in sheep (Fig. 1a) and generated multiple layers of data (Fig. 1b), including whole genome sequences (WGS), whole-body transcriptomes, chromatin accessibility (i.e., ATAC-Seq) and single-cell RNA-Seq (scRNA-Seq) data, and blood physiological and biochemical phenotypes. We aimed to (1) identify time-series expression changes and regulatory elements in response to short-term hypoxia across tissues; (2) reveal multi-tissue expression patterns of genes implicated in high-altitude adaptation and diseases in human; and (3) test the acclimatization of offspring to hypoxia. This research promotes our understanding of the mechanisms conferring resilience to hypoxia and provides valuable resources for future studies of hypoxia-related diseases in human and livestock.

## Results

### Low-to-high altitude translocation experiment

Hu sheep and Tibetan sheep, two representative Chinese native breeds that originally inhabited low-altitude plain (Zhejiang Province, China) and the high-altitude Qinghai-Tibet Plateau (QTP), respectively, were included in the experiment. Tibetan sheep initially spread to the QTP from northern China along with the colonization of nomads ~3100 years ago and have become well adapted to the high-altitude environment[29]. There were three different scenarios in our experiment (Fig. 1a): (i) scenario 1, low- altitude Hu sheep (i.e., Hu sheep ewes from low altitude, $n = 10$) raised in Neijiang (~350 m.a.s.l, the southeast of Sichuan Province, "0 d" hereafter); (ii) scenario 2, high- altitude Hu sheep ($n = 43$: ewes, $n = 40$; rams, $n = 3$) that were translocated from the low altitude to Aba Autonomous Prefecture (~3500 m.a.s.l, eastern edge of the QTP in Sichuan Province) and acclimatized for four different time periods after translocation (i.e., 7 days, 14 days, 21 days and ~8 months, "7 d, 14 d, 21 d, 8 mon" hereafter); and (iii) scenario 3, Tibetan sheep (i.e., Tibetan sheep ewes at high altitude; $n = 10$) raised in Aba. In addition, offspring ($n = 18$) of the ewes under the above three scenarios (six lambs for each scenario) were included in our experiment.

### Data summary

To comprehensively study transcriptomic, epigenomic and phenotypic changes in sheep acclimatization from the low altitude to the high altitude, we collected multiple tissues from experimental animals to perform high-throughput sequencing (Fig. 1b). We produced 49 WGS from heart tissue and 1277 RNA-Seq datasets from 19 major tissues (Supplementary Data 1 and 2). We then uniformly processed the data, yielding ~24 billion uniquely mapped paired-end reads with an average mapping rate of 97.92% (95.89 – 98.31%) for the WGS datasets (Supplementary Data 3) and ~23 billion reads with an average mapping rate of 84.3% (61.05 – 91.37%) for the RNA-Seq datasets (Supplementary Data 4). We also generated 66 chromatin accessibility datasets for eight tissues (i.e., heart, artery, lung, liver, hypothalamus, rumen, duodenum, and adipose) by ATAC-Seq, and produced six high-resolution scRNA-Seq datasets for lung tissue (i.e., for low-altitude Hu sheep, high-altitude Hu sheep at four acclimatization time points and Tibetan sheep) (Supplementary Data 1 and 2). After raw data processing, we obtained ~4.5 billion informative reads with an average unique mapping rate of 98.63% (84.57 – 99.27%) for ATAC-Seq (Supplementary Data 5) and ~ 4.0 billion reads with a confident mapping rate of 79.71% (52.70 – 95.40%) for scRNA-Seq (Supplementary Data 6).

To evaluate physiological acclimatization under high-altitude hypoxia, we collected phenotypic data for 20 blood parameters, including blood oxygen saturation ($SpO_2$) and 19 bio-indicators (e.g., erythropoietin, nitric oxide, and cardiac enzymes) (Supplementary Data 7 and 8). We also included 37 whole-genome sequences[30] and high-throughput chromosome conformation capture (Hi-C) data from a sheep that were previously published[31] for the integrated analysis (Fig. 1c and Supplementary Data 9 and 10).

### Phenotypic and transcriptional characteristics

Previous evidence from several vertebrate taxa suggested that physiological adjustments have played a significant role in high-altitude hypoxia tolerance and could well represent environment-induced physiological changes[3,32]. We examined 10 Hu sheep ewes before (i.e., 0 d) and after their translocation to the QTP at four time points (i.e., 7 d, 14 d, 21 d and ~8 mon) to track the changes in $SpO_2$ and other blood indicators. We observed that the mean value of $SpO_2$ decreased sharply (0 d vs. 7 d, 0 d: 97.3, 95% confidence interval: 96.68 – 97.92; 7 d: 82.5, 95% confidence interval: 81.87 – 83.15; Wilcoxon rank sum test, $P = 7.50 \times 10^{-13}$) at 7 d but increased constantly at 14 d and later with acclimatization (Fig. 2a and Supplementary Data 11). However, compared to Tibetan sheep, Hu sheep still exhibited significantly lower levels of $SpO_2$ (Hu sheep: 87, 95% confidence interval: 96.68 – 97.92; Tibetan sheep: 92.65, 95% confidence interval: 91.68 – 93.62; Wilcoxon rank sum test, $P = 1.40 \times 10^{-9}$) even after 8 months (Fig. 2a and Supplementary Data 11), probably reflecting potentially different mechanisms underlying phenotypic plasticity and genetic adaptation to high-altitude hypoxia in mammals[3]. In addition to $SpO_2$, we observed distinct patterns of the other bioindicators in the translocated Hu sheep (Supplementary Fig. 1). For example, nitric oxide synthetase (NOS), which restricts the synthesis of the vasodilator NO, decreased after translocation and showed significantly lower average values than in Tibetan sheep (Hu sheep: 5.43 μmol/L, 95% confidence interval: 6.69 – 7.41; Tibetan sheep: 6.40 μmol/L, 95% confidence interval: 5.88 – 6.92; Wilcoxon rank sum test, $P = 0.013$) (Fig. 2b and Supplementary Data 11), verifying that down-regulated NO synthesis contributes to hypoxic pulmonary vasoconstriction[32–34]. In general, the levels of triglycerides (TG) and glucose (GLU), which are associated with energy metabolism, increased over time (Supplementary Fig. 1 and Supplementary Data 11), suggesting enhanced energy production in response to hypoxia. The change in total bilirubin (T-BIL), an indicator that is positively correlated with liver damage, followed a bell curve (Supplementary Fig. 1 and Supplementary Data 11), implying that liver impairment was gradually relieved during acclimatization. Cardiac enzymes (CK) showed non-significant changes among the four-time points examined during acclimatization ($P > 0.05$) but presented significantly lower averages than in Tibetan sheep (Hu sheep: 58.96 U/L, 95% confidence interval: 48.64 – 75.56; Tibetan sheep: 146.7 U/L, 95% confidence interval: 69.15 – 224.25; Wilcoxon rank sum test, $P = 0.0087$) (Fig. 2c and Supplementary Data 11), indicating non-activation during short-term hypoxia.

To characterize the global expression patterns of whole-body tissues, we first applied *t*-distributed stochastic neighbour embedding (*t*-SNE) analysis based on the gene expression profiles of samples. The

## a Animal experiment

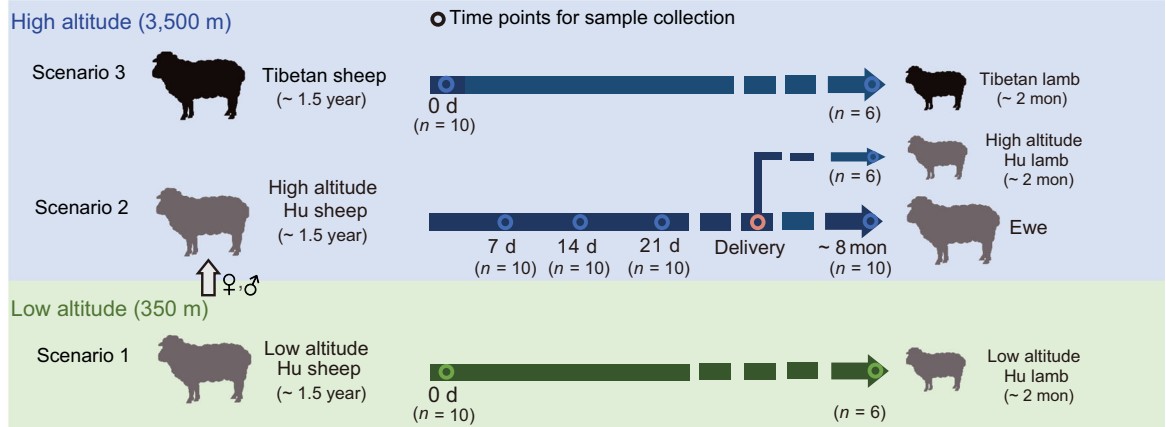

## b Data generation

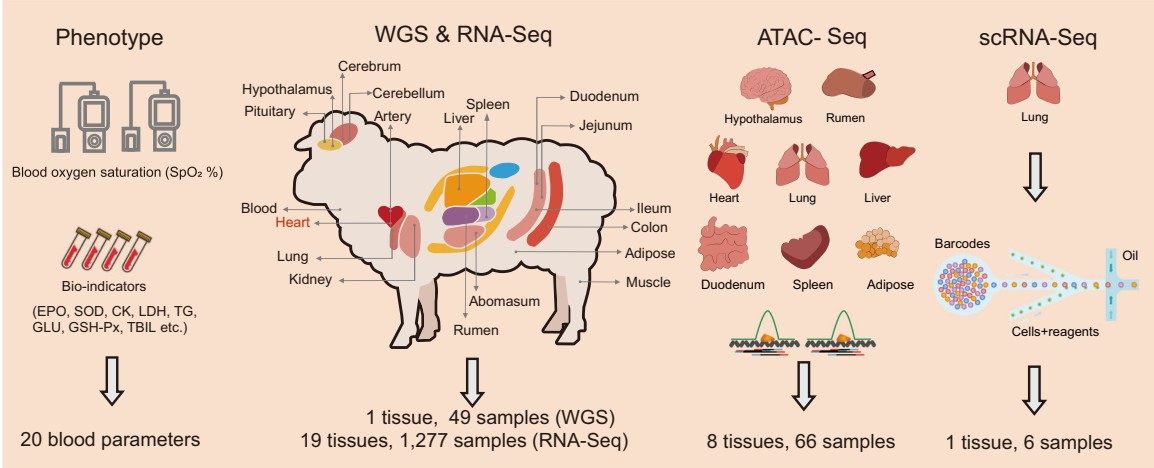

## c Bioinformatics analysis

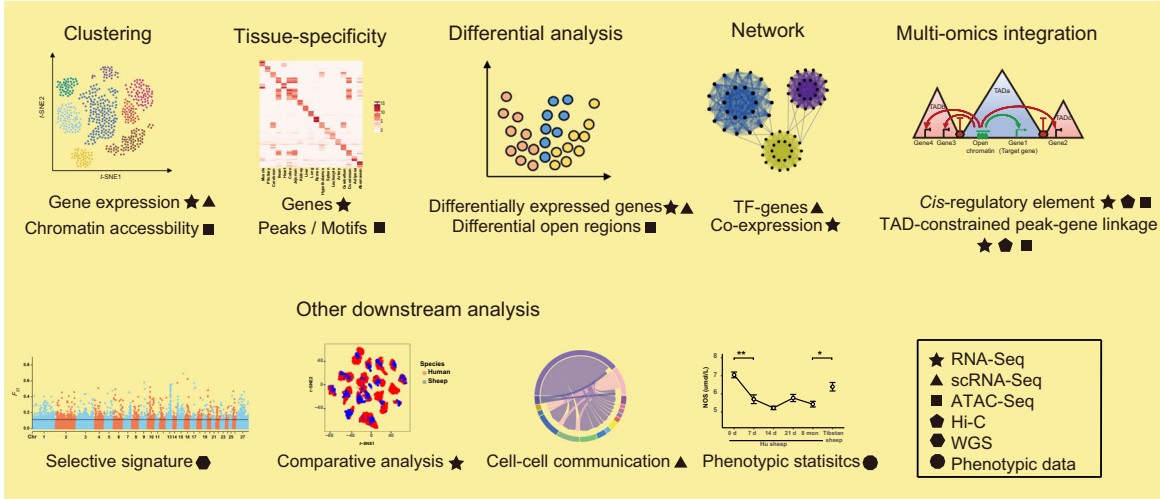

**Fig. 1 | Schematic diagram of the study. a** Design of the animal translocation experiment. Hu sheep (gray) and Tibetan sheep (black) originally inhabited low-altitude and high-altitude environments, respectively. There were three scenarios examined in our experiment: low-altitude Hu sheep raised in the lowlands (scenario 1); high-altitude Hu sheep, namely low-altitude Hu sheep that were translocated to the highlands (scenario 2) and acclimatized until four time points; and Tibetan sheep raised in highlands (scenario 3). In addition, offspring of the ewes in the above three scenarios were included. **b** Sample collection and data generation. We collected 19 whole-body tissues and produced phenotypic, genomic (WGS), transcriptomic (bulk-RNA, single-cell RNA), and epigenetic (ATAC-Seq) data. Tissue marked in red (i.e., heart) was used for generating WGS data. **c** Major bioinformatics and statistical analysis involved in the study.

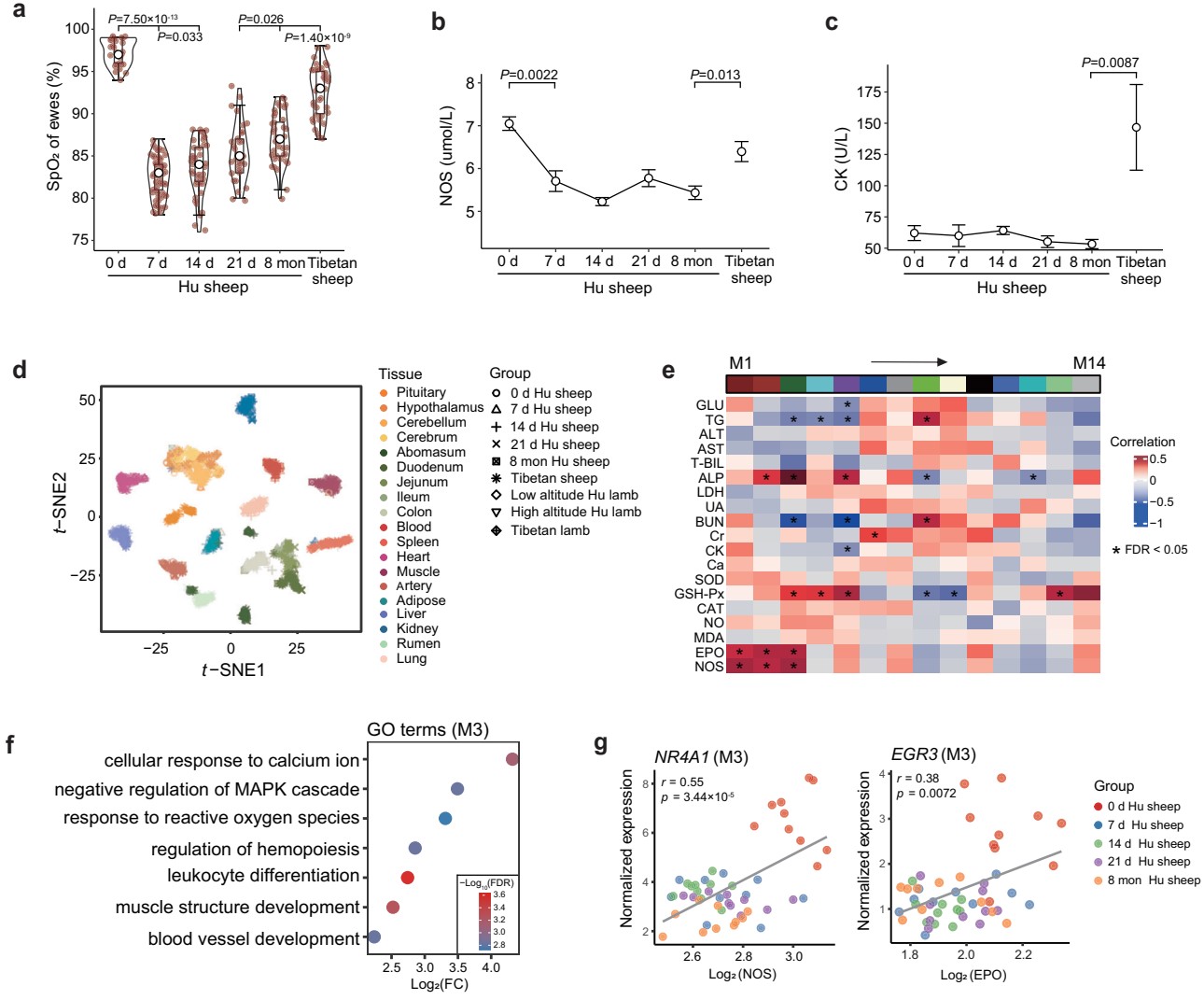

**Fig. 2 | Characteristics of phenotype and gene expression. a** Changes in blood oxygen saturation [SpO₂, average value with 95% confidence interval (CI)] with acclimatization time. Hu sheep (*n* = 10): 0 d, 97.30 (CI: 96.68 − 97.92); 7 d, 82.51 (CI: 81.87 − 83.15); 14 d, 83.67 (CI: 82.68 − 84.66); 21 d, 85.38 (CI: 84.24 − 86.52); 8 mon, 87.0 (CI: 86.03 − 87.97); and Tibetan sheep (*n* = 10), 92.65 (CI: 91.68 − 93.62). Boxplots are represented by minima, 25% quantile, median, 75% quantile, and maxima with data points. **b, c** Changes in the nitric oxide synthetase (NOS) (b) and cardiac enzyme (CK) (c) values (average value with 95% CI) over time. In **b**, Hu sheep (*n* = 10): 0 d, 7.05 μmol/L (CI: 6.69 − 7.41); 7 d, 5.71 μmol/L (CI: 5.17 − 6.25); 14 d, 5.23 μmol/L (CI: 5.03 − 5.43); 21 d, 5.78 μmol/L (CI: 5.33 − 6.23); 8 mon, 5.43 μmol/L (CI: 5.07 − 5.79); and Tibetan sheep (*n* = 10), 6.40 μmol/L (CI: 5.88 − 6.92). In **c**, Hu sheep (*n* = 10): 0 d, 62.1 U/L (CI: 48.64 − 75.56); 7 d, 60.0 U/L (CI: 40.45 − 79.55); 14 d,

64.2 U/L (CI: 56.87 − 71.53); 21 d, 55.2 U/L (CI: 44.64 − 65.76); 8 mon, 53.3 U/L (CI: 44.91 − 61.69); and Tibetan sheep (*n* = 10), 146.7 U/L (CI: 69.15 − 224.25). *P* values in the figures **a**−**c** come from the two-sided Wilcoxon rank sum test. **d** *t*-distributed stochastic neighbor embedding (*t*-SNE) clustering of 1277 RNA-Seq samples. **e** Association between gene modules and bio-indicators in blood. The rows represent the 14 gene modules (i.e., M1-M14), and the columns show 19 bio-indicators. Multiple testing was corrected using the Benjamini-Hochberg method. * FDR < 0.05. **f** Gene Ontology (GO) analysis for M3. **g** Gene examples in M3. Scatter plots show the Pearson's correlation between the expression levels of genes and the values of bio-indicators over time. The two-sided *P* values are calculated by the linear regression model. Source Data are provided as Source Data file.

resultant sample clustering recapitulated the different tissues accurately (Fig. 2d), consistent with the hierarchical clustering of expression profiles (Supplementary Fig. 2a–d). The functions of the genes with tissue-specific expression reflected known tissue biology (Supplementary Data 12). Next, we explored the effect of hypoxia on gene expression within tissues over time. We only observed a particular pattern of gene expression in the abomasum, which showed an obvious separation of samples before and after 14 d (Supplementary Fig. 3). In particular, *GKN2*, an abomasum-specific gene that is associated with oxidative stress-induced gastric cancer cell apoptosis[35,36], showed increased expression with time (Supplementary Fig. 4).

Furthermore, we investigated associations between blood gene expression and bio-indicators using weighted correlation network analysis (WGCNA). Among 13,707 genes remaining after filtration, we

determined 14 gene modules (in M1-M14) labeled with different colors (Supplementary Fig. 5a, b), and 10 gene modules were significantly correlated (FDR < 0.05) with one or more bio-indicators (Fig. 2e, Supplementary Fig. 5c and Supplementary Data 13). In particular, we observed that M3 was significantly positively associated with erythropoietin (EPO), nitric oxide synthetase (NOS), glutathione peroxidase (GSH-Px) and alkaline phosphatase (ALP). The results of Gene Ontology (GO) enrichment among the genes in M3 were congruent with their associated bio-indicators (Fig. 2f). For example, EPO is involved in the regulation of hemopoiesis (e.g., *EGR3*, *HOX5* and *FOS*), NOS is associated with blood vessel development (e.g., *NR4A1*, *JUN* and *RHOB*), and GSH-Px is related to the response to reactive oxygen species (e.g., *NR4A3*, *PLK3* and *TNFAIP3*). Protein-protein interaction analysis also showed that genes (e.g., *JUN*, *FOS* and *NR4A1*) related to

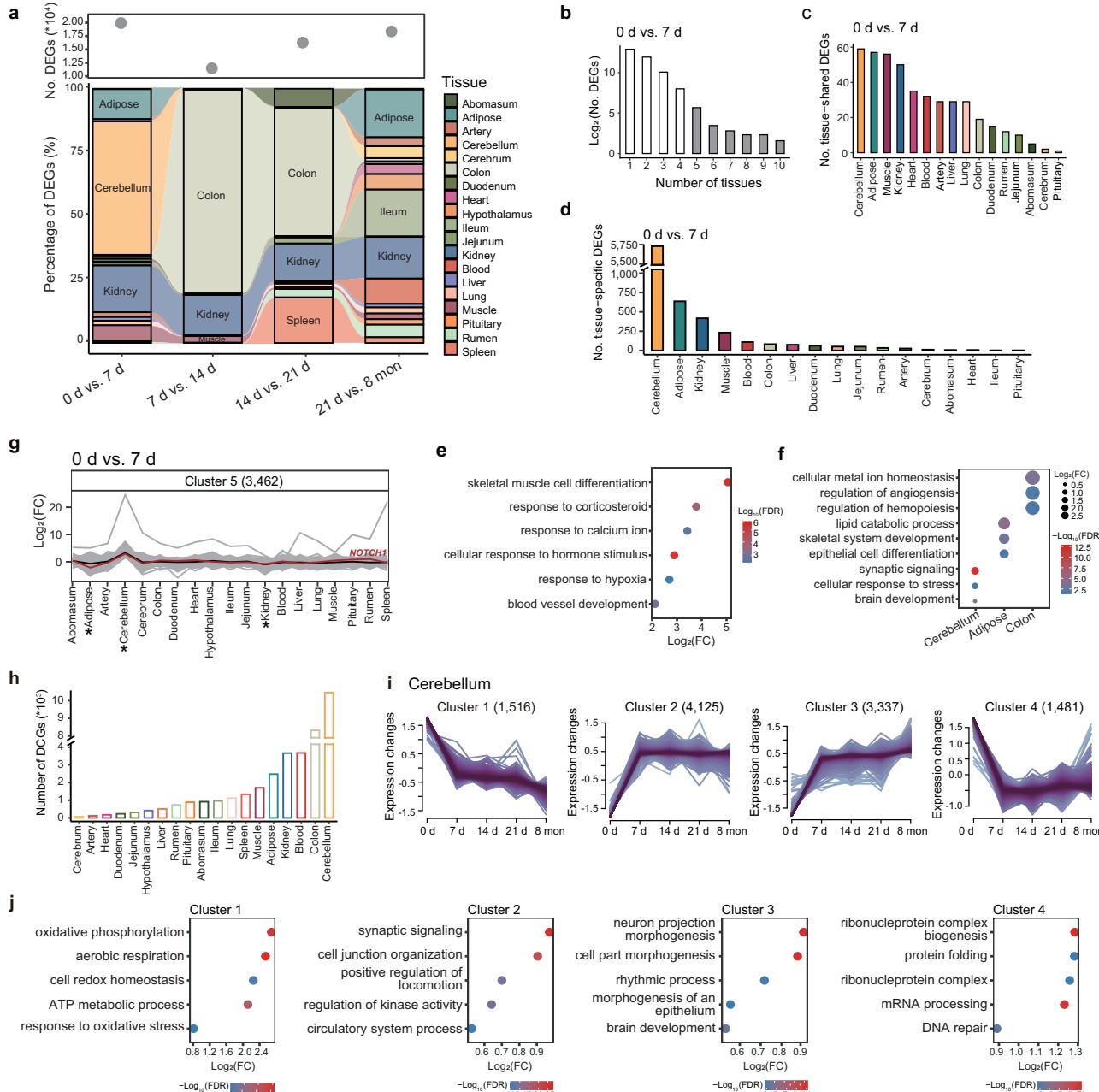

**Fig. 3 | Transcriptome dynamics during hypoxia acclimatization. a** Numbers of differentially expressed genes (DEGs) (top) and percentages of DEGs (bottom) between adjacent time points comparisons across tissues. **b** Distribution of DEGs across numerous tissues in the "0 d vs. 7 d" comparison. **c**, **d** Numbers of tissue-shared (**c**) and tissue-specific (**d**) DEGs across tissues in the "0 d vs. 7 d" comparison. **e**, **f** GO term enrichments for tissue-shared (**e**) and tissue-specific (**f**) DEGs from the "0 d vs. 7 d" comparison. **g** Multi-tissue interactions in the "0 d vs. 7 d" comparison.

The average log$_2$FC value for cluster 5 is denoted with a black line. Active tissues (i.e., cerebellum, kidney, and colon) are marked with asterisks, and the interaction of *NOTCH1* between tissues is highlighted with a red line. **h** Numbers of dynamically changed genes (DCGs) across tissues. **i** Fuzzy *c*-means clustering identified gene expression patterns of DCGs in the cerebellum. **j** GO terms for the four clusters identified in (**i**). Source Data are provided as Source Data file.

the above GO terms were at the center of the regulatory network within the genes in M3 (Supplementary Fig. 5d). Furthermore, we explored the changes between gene expression and bio-indicators over time (Supplementary Fig. 6). The overall expression levels of *NR4A1* and *EGR3* (from gene module M3 in Fig. 2e) were significantly and positively correlated with NOS (Pearson's $r = 0.55$, $P = 3.44 \times 10^{-5}$) (Fig. 2g) and with EPO (Pearson's $r = 0.38$, $P = 0.0072$) (Fig. 2g), respectively. *NR4A1* showed higher expression and higher NOS values at 0 d (i.e., normoxia) (Supplementary Fig. 5e). A similar expression trend was also observed for *EGR3* (Supplementary Fig. 5f).

**Temporal transcriptome dynamic and multi-tissue interaction**

To explore the transcriptional changes during hypoxia acclimatization, we identified differentially expressed genes (DEGs) between five adjacent time points across tissues. Overall, we observed large variations in transcriptional regulation between and within tissues in terms of the number of DEGs, particularly between 0 d and 7 d (Fig. 3a). We identified the most active tissues (i.e., tissues with the top 3 counts of DEGs) in each comparison. Certain tissues (e.g., kidney, colon, adipose and cerebellum) showed activation in the comparisons of multiple or, particularly, adjacent time points (Fig. 3a). We performed power

analyzes to test the credibility of differential expression analyzes in comparison of "0 d vs. 7 d" in cerebellum and "7 d vs. 14 d" in colon. The results showed that FDR of the actual detection of DEGs was less than 0.05 when we used the same thresholds (i.e., |log$_2$FC| > 0.75) for simulated data and actual data (Supplementary Fig. 7), indicating that a large number of DEGs identified in the active tissues are reliable and not technical artifact. For example, kidney showed activation in all four comparisons of adjacent time points, indicating that kidney functions concerning ATP production and stress hormone secretion were important for the hypoxia response[37,38]. Colon showed activation in both the "7 d vs. 14 d" and "14 d vs. 21 d" comparisons, implying that hypoxia strongly affects intestinal homeostasis[39] during acclimatization. We also found that the cerebellum only showed activation in "0 d vs. 7 d", which suggested that hypoxia severely affects the cerebellum first, before the other examined tissues. These observations demonstrated that certain organs or tissues, such as kidney, colon, adipose, and cerebellum, that were actively involved in rapid hypoxia acclimatization had main functions (e.g., energy metabolism, endocrine, and nervous system functions) that differed from those of tissues (e.g., heart and lung) involved in long-term hypoxia adaptation, such as cardiovascular and respiratory functions.

To further dissect the magnitude of transcriptional change during hypoxia acclimatization, we also identified DEGs across tissues in high-altitude Hu sheep at four-time points (7 d, 14 d, 21 d, and 8 mon) in comparison with low-altitude Hu sheep (0 d) (Supplementary Data 14). Similar to the adjacent time comparison, we detected a large number of DEGs in cerebellum in all comparisons and in colon from "0 d vs. 14 d" comparison (Supplementary Fig. 8). Power analyzes evaluated false discovery rate (FDR) of the identified DEGs in the above tissues and found the FDR values were less than 0.05, which support high credibility and exclude technical artifacts in these DEG detection (Supplementary Figs. 9 and 10).

Next, we examined the distribution of DEGs across tissues for each comparison between adjacent time points. Most of the DEGs were assigned to particular tissues, while a small number of DEGs showed a ubiquitous distribution (Fig. 3b and Supplementary Data. 11). We then identified tissue-shared (i.e., in at least five tissues) and tissue-specific DEGs (i.e., in only one tissue) for each comparison (Fig. 3c, d and Supplementary Data 15 and 16). In the comparison of "0 d vs. 7 d", the functional enrichment of tissue-shared DEGs showed the involvement of the genes in multiple biological processes (Supplementary Data 17), such as skeletal muscle cell differentiation (e.g., *BTG2*, *ATF3*, and *NR4A1*), response to hypoxia (e.g., *NR4A2*, *EGR1*, and *CPEB2*) and the response to corticosteroids (e.g., *NR4A3*, *IGF1R*, and *FOS*) (Fig. 3e). However, tissue-shared DEGs from the other three comparisons mostly participated in energy metabolism, such as mitochondrial organization (e.g., *NDUFAF8*, *ROMO1*, and *UQCC2*) and ATP metabolic processes (e.g., *ND1*, *COX2* and *ATP5PF*) (Supplementary Data 17). These results implied that the initial hypoxic stimulus (e.g., the first 7 d after translocation to QTP) resulted in collective responses of multiple life systems[4]. Additionally, the functions of tissue-specific DEGs reflected the respective tissue biology and hypoxia response (Supplementary Data 17). For instance, in the comparison of "0 d vs. 7 d", cerebellum-specific DEGs were associated with synaptic signaling (e.g., *NTNG1*, *TNF*, and *GABBR2*) and regulation of the cellular response to stress (*HIF1A*, *ATF4*, and *PARG*), while colon-specific DEGs were involved in cellular metal ion homeostasis (e.g., *HMOX1*, *TRPM8*, and *CXCR5*) and the regulation of angiogenesis (e.g., *ISL1*, *THBS4*, and *ADGRB2*) (Fig. 3f). Taken together, the findings indicated that hypoxia acclimatization could have activated both hypoxia response processes in multiple tissues and the functions of specific tissues, which were collectively regulated by polygenic (i.e., tissue-specific DEGs) and pleiotropic (i.e., tissue-shared DEGs) genes[40].

Previous evidence suggests that the maintenance of systemic homeostasis and responses to environmental challenges typically requires transcriptional interactions among multiple organs and tissues[41,42]. To identify transcriptional interactions underlying hypoxia acclimatization, we used the *k*-means method to analyze multi-tissue interactions based on log$_2$FC (log$_2$-transformed fold change) values in the above differential expression analysis and obtained 5 – 7 clusters in different comparisons (Supplementary Fig. 12a–d). Overall, in the comparison of "0 d and 7 d", gene cluster 5 (e.g., *NOTCH1*) showed the greatest increase in expression in the cerebellum but decreased expression in kidney and adipose at 7 d compared to 0 d (Fig. 3g and Supplementary Fig. 12e). Notably, we observed possible interactions among the most active tissues, such as cerebellum, kidney and adipose in the "0 d vs. 7 d" comparison (Fig. 3a) and colon in the "7 d vs. 14 d" and "14 d vs. 21 d" comparisons (Supplementary Fig. 12b, c). These findings implied the action of potential transcriptional networks among particular tissues at different time points during acclimatization.

## High sensitivity of cerebellum in response to hypoxia

To explore the expression patterns within tissues across time points, we conducted a time-series differential expression analysis to identify dynamically changed genes (DCGs) (i.e., genes with significant expression changes throughout the acclimatization process). Since genes with similar expression patterns could be involved in the same biological process[43,44], we further classified DCGs into different gene clusters based on their expression patterns with the *c*-means method[45] (Supplementary Data 18). The numbers of our DCGs in different tissues ranged from 68 (cerebrum) to 10,459 (cerebellum) (Fig. 3h) and were categorized into 2 – 6 clusters across tissues (Supplementary Fig. 13). In most tissues, the changes in the expression of DCGs over time reflected similar patterns of the temporal transcriptional changes described above (Fig. 3a and Supplementary Fig. 13). For example, DCGs in the cerebellum were categorized into four clusters (Fig. 3i), and the overall expression patterns of these clusters varied greatly at 7 d. This observation was consistent with the large transcriptional changes in the cerebellum in the "0 d vs. 7 d" comparison (Fig. 3a). Furthermore, the DCGs in each cluster from the cerebellum exhibited distinct biological functions (Fig. 3j). Specifically, the functions of the DCGs in cluster 1 were associated with energy metabolism (e.g., aerobic respiration and ATP metabolic process), while in cluster 3, the gene functions were related to brain biology (e.g., neuron projection morphogenesis and brain development) (Fig. 3j). These results revealed adjustments in energy metabolism and the biological function of the cerebellum in response to hypoxic challenge.

## Hypoxia-adaptive genes in adaptation and acclimatization

The evolution of gene expression and regulation is a major source of phenotypic diversity[46–48]. To explore the roles of hypoxia-adaptive genes in long-term adaptation and short-term acclimatization, we integrated gene expression data with genome sequencing data to perform a joint analysis. We first calculated pairwise $F_{ST}$ values between 37 genomes of 7 sheep breeds originating from the low altitude (*n* = 20) and high altitude (*n* = 17) regions and chose the top 5% of the $F_{ST}$ distribution as candidate selected regions (Supplementary Fig. 14a and Supplementary Data 19). The functional annotation of putatively selected genes (i.e., $F_{ST}$ genes) from the candidate regions revealed their high relevance to high-altitude adaptation (Supplementary Fig. 14b and Supplementary Data 20). We detected DEGs between low-altitude Hu sheep and Tibetan sheep in each of the tissues (Supplementary Data 21). For each tissue, we intersected $F_{ST}$ genes with inter-breed DEGs and DCGs separately. We examined the distribution of two categories of intersected genes (i.e., $F_{ST}$ genes in DEGs and $F_{ST}$ genes in DCGs) across tissues (Fig. 4a, Supplementary Data 22 and Supplementary Fig. 15). We identified 52 multi-tissue $F_{ST}$ genes (i.e., $F_{ST}$ genes in at least five tissues) in DEGs and 179 multi-tissue $F_{ST}$ genes in DCGs (Supplementary Data 22 and 23), including 13 common

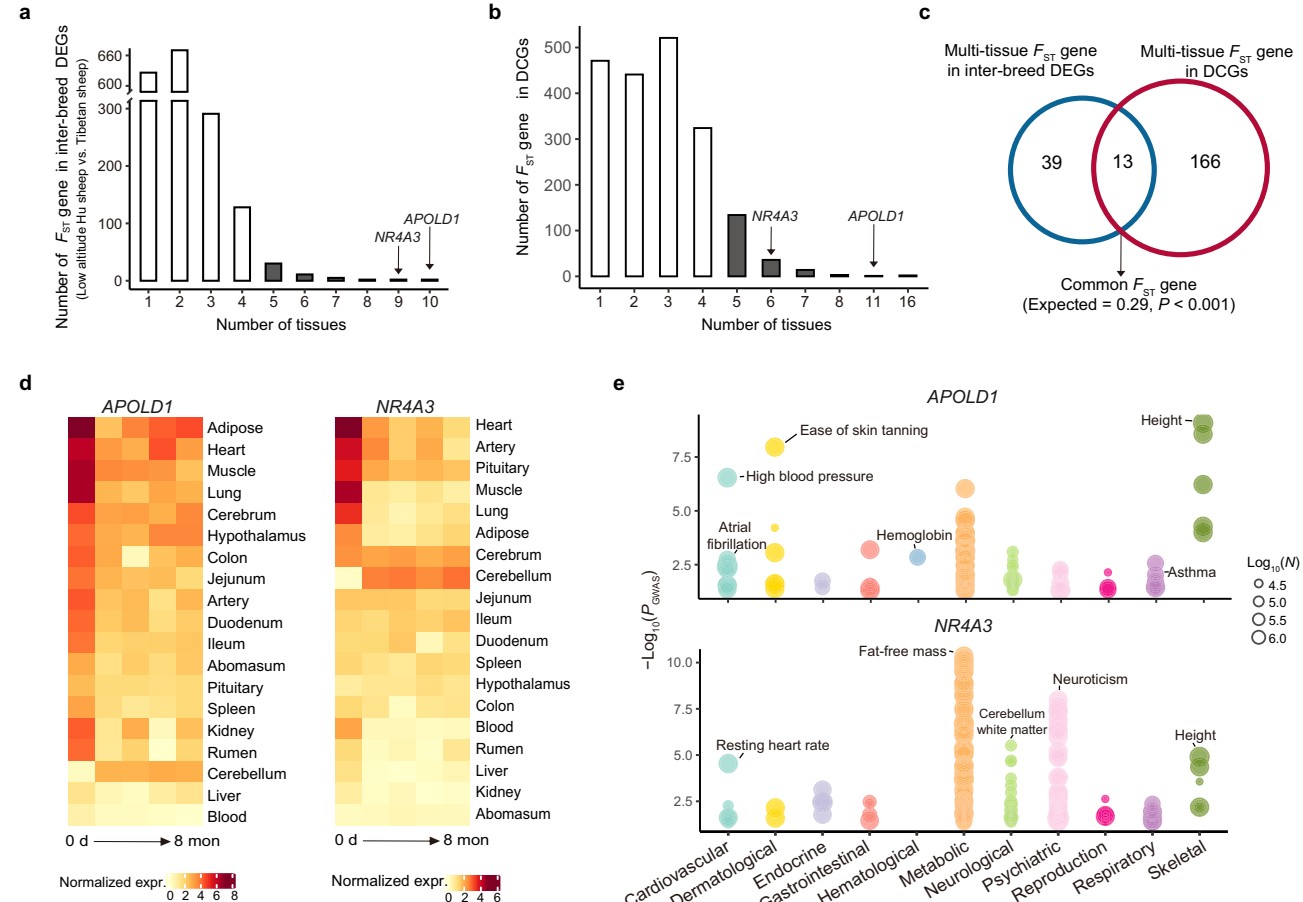

**Fig. 4 | Hypoxia-adaptive genes in adaptation and acclimatization.**
**a** Distribution of $F_{ST}$ genes among differentially expressed genes (DEGs) between breeds (i.e., low-altitude Hu sheep vs. Tibetan sheep) across tissues. **b** Distribution of $F_{ST}$ genes in DCGs across tissues. **c** Venn diagram showing the intersection of multi-tissue $F_{ST}$ genes in inter-breed DEGs with those in DCGs. *P* value is calculated by two-sided permutation test. d-e, Examples of common multi-tissue $F_{ST}$ genes.

*P* value was calculated by permutation test with 1000 times shuffle. **d** Expression changes in *APOLD1* (left) and *NR4A3* (right) over time across tissues. expr, expression. **e** Phenome-wide association analysis (Phe-WAS) for *APOLD1* (top) and *NR4A3* (bottom). *N* is the sample size of GWAS. *P* values are calculated by two-sided Chi-square test and multiple correction is used Benjamini-Hochberg method. Source Data are provided as Source Data file.

genes (i.e., *APOLD1*, *NDUFB9*, *ERBB4*, *NFKBIZ*, *NR4A3*, *RPS8*, *CIAO2A*, *AHCYL2*, *ESRRG*, *KIAAO930*, *RASGEF1B*, *MRPS25* and *TNFRSF21*) (Fig. 4c and Supplementary Data 24). The 13 common genes are significantly greater than expected by chance (permutation test, *P* < 0.001) (Fig. 4c). The results indicated that these 13 $F_{ST}$ genes could have played an important role in hypoxia adaptation and acclimatization by regulating the expression of multiple tissues. Furthermore, we examined changes in the expression levels of these genes across sheep tissues (Fig. 4d and Supplementary Fig. 16) and investigated their functions in the human GWAS atlas database (Supplementary Data 25). We found that the human trait/disorder associations (e.g., hypoxia-related traits) of these genes were largely consistent with dynamic expression changes in analogous sheep tissues. For instance, *APOLD1*, which was significantly (*P* < 0.05) associated with cardiovascular (e.g., high blood pressure), respiratory (e.g., asthma) and hematological (e.g., hemoglobin) traits (Fig. 4e and Supplementary Data 25), showed significant expression changes in the heart, lung and kidney (Fig. 4d). Likewise, *NR4A3* was significantly (*P* < 0.05) associated with metabolic (e.g., fat-free mass), nervous/neurological (e.g., neuroticism and insomnia) and cardiovascular (e.g., resting heart rate) traits (Fig. 4e and Supplementary Data 25) and was dynamically expressed in adipose, cerebellum, heart and artery (Fig. 4d). These results suggested that the 13 identified tissue-shared hypoxia-adaptive genes could regulate hypoxia-related traits by controlling expression in different tissues in both genetic adaptation and short-term acclimatization.

Additionally, we conducted permutation test to examine whether the overlaps between $F_{ST}$ genes and all DEGs within Hu sheep from 19 tissues were significantly higher than that expected at random. The result showed that 1770 overlapped genes between long-term adaptation (i.e., 2648 $F_{ST}$ genes) and short-term acclimatization (i.e., 13,891 DEGs) is significantly higher than that expected at random (i.e., 1234 genes, *P* < 0.001) (Supplementary Fig. 15), implying that long-term selection may favor individuals with SNPs that correspond to DEGs during short-term acclimatization.

**Chromatin accessibility across tissues under hypoxia**
To identify regulatory elements related to dynamic expression, we applied ATAC-Seq to detect genomic chromatin accessibility across eight important tissues (i.e., hypothalamus, rumen, heart, lung, liver, duodenum, spleen and adipose) under the three scenarios described above. We obtained a total of 1,662,152 statistically significant peaks (*P* < 0.01) (Supplementary Fig. 17a), and the open chromatin regions in the eight tissues were highly enriched at transcription start site (TSS) (Fig. 5a). The distribution of peaks varied among tissues (Supplementary Fig. 17b), but in general, the highest proportion of peaks were located in intergenic regions, followed by intronic region, while the lowest proportion were located in 5' UTRs (Supplementary Fig. 17c). One exception was adipose, which showed the largest number of peaks in regions ≤ 1 kb from promoters. The ATAC-Seq signals showed strong correlations among biological replicates at the whole-genome level

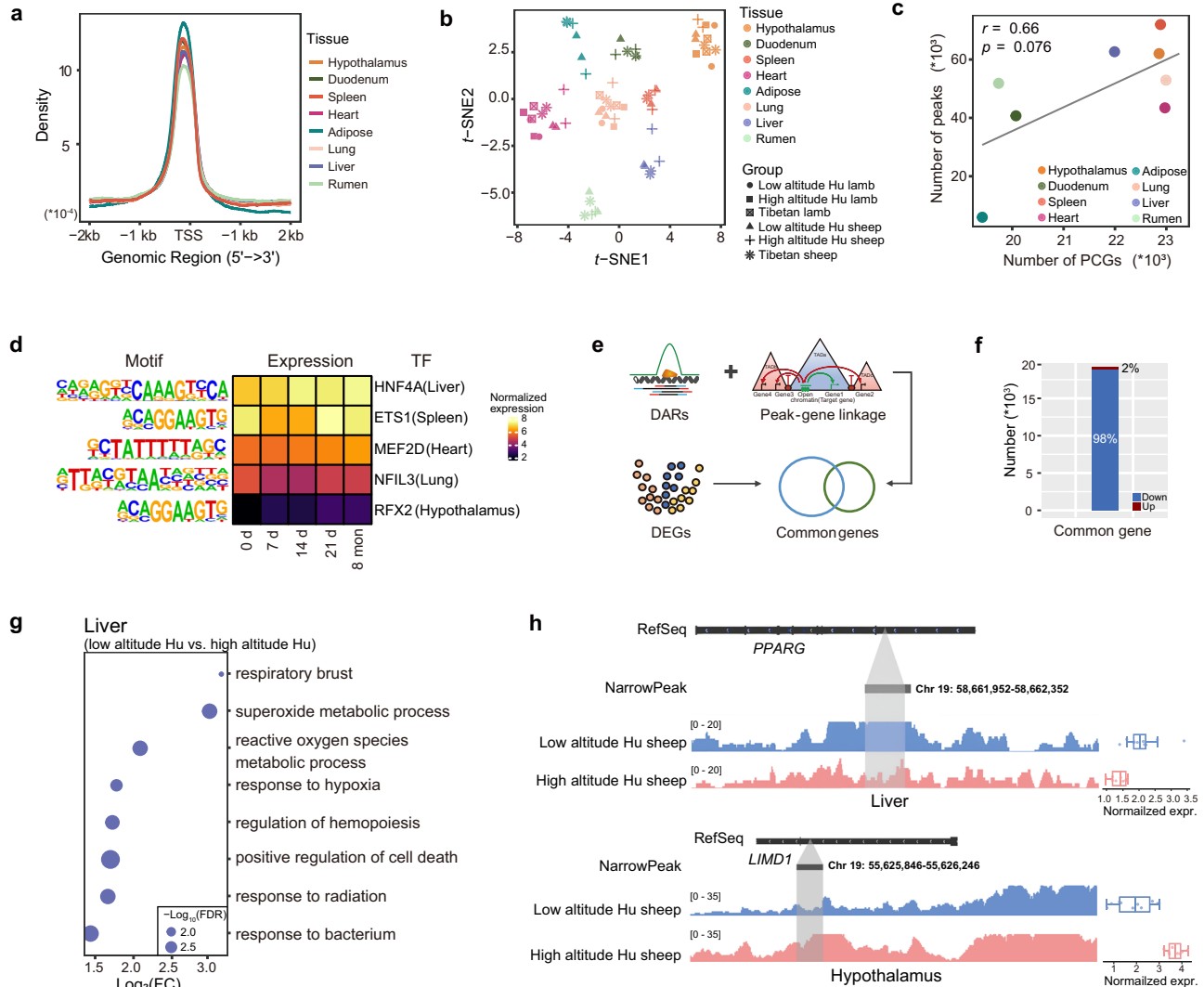

**Fig. 5 | Chromatin accessibility reveals the regulatory landscape of hypoxia acclimatization. a** Average peak density of each tissue at positions relative to transcription start site (TSS). **b** *t*-SNE clustering of 66 samples based on peak signal density. **c**, Pearson's correlations between the numbers of expressed genes and detected peaks across tissues. The two-sided *P* values are calculated by the linear regression model. PCGs: protein coding genes. **d** Tissue-specific transcription factors (TFs) in differentially accessible regions (DARs) of the respective tissues and their gene expression over time. **e** Identification of common genes. **f** Numbers of up-regulated and down-regulated peak-gene pairs. **g** Common genes of liver from

low-altitude Hu sheep vs. high-altitude Hu sheep comparison are enriched in biological processes related to the hypoxia response. **h** Examples of common up-regulated and down-regulated genes (*n* = 6). Peak density and expression level of the down-regulated common gene *PPARG* (top) in the liver and the up-regulated common gene *LIMD1* (bottom) in the hypothalamus from low-altitude Hu sheep and high-altitude Hu sheep (8 mon) comparison. RefSeq, reference sequence. expr, expression. Boxplots are represented by minima, 25% quantile, median, 75% quantile, and maxima. Each dot represents individual expression level in different groups.

(Supplementary Fig. 17d) and were clustered by different tissues instead of breeds and time points (Fig. 5b). Moreover, we observed a strong positive correlation (Pearson's *r* = 0.66, *P* = 0.076) between the numbers of protein-coding genes (PCGs) and ATAC-Seq peaks across tissues (Fig. 5c), which demonstrated that open chromatin regions positively regulate transcriptional activity.

As hotspots for transcription factor (TF) activities, open chromatin landscapes have unique effects in driving the biological functions of tissues[49,50]. We characterized tissue-specific and conserved peaks across tissues. The results showed that peaks located in the distal intergenic and intron regions are more tissue-specific, whereas those in the promoter, exon, 3′ UTR, 5′ UTR and downstream regions are more conserved (Supplementary Fig. 18a, b). We further identified corresponding tissue-specific TFs (Supplementary Data 26), and TF binding motifs (TFBMs) were significantly (*P* < 0.05) enriched in the specific peaks of various tissues, such as the TFBMs of MEF2D in heart,

HNF4A in liver and ETS1 in spleen (Supplementary Fig. 18c, d). We also implemented differential expression analysis between pairwise comparisons of low-altitude Hu sheep, high-altitude Hu sheep after translocation to the QTP for 8 months, and Tibetan sheep for each tissue and determined TFBMs in differentially accessible regions (DARs) (Supplementary Data 27). We found that some tissue-specific TFBMs were significantly (*P* < 0.05) enriched in DARs of corresponding tissues (Fig. 5d). For example, the heart-specific TFBM of MEF2D was found in DARs of heart tissue. The expression of *MEF2D* gradually increased with time (Fig. 5d), suggesting the continuous activation of MEF2D target genes in the heart, which is in line with the role of *MEF2D* in the regulation of cardiac muscle[51].

## Regulation of gene expression by cis-regulatory elements

To leverage the chromatin accessibility information captured by integrating gene expression, we used a correlation-based method to

predict *cis*-regulatory elements (CREs) and their target genes within the same topologically associated domains (TADs), enabling the capture of all CREs (e.g., promoters and enhancers). We obtained 3032 TADs using previously published Hi-C data which was generated from the blood of Tibetan sheep[31]. A total of 2,875,658 independent peak-gene assignments were derived from all the TADs, and after filtration, 460,421 high-quality peak-gene pairs were retained for the following analysis. Subsequently, we examined the functions of target genes for the tissue-specific and conserved peaks. The target genes reflected tissue specificity and tissue biological functions well (Supplementary Data 28). For example, lung target genes were significantly (FDR < 0.05) associated with lung development (e.g., *FOXF1*, *NKX2-1*, and *LIF*) and epithelial cell differentiation (e.g., *HOXA7*, *TMOD1*, and *SOX17*) (Supplementary Fig. 18e). This observation indicated that CREs frequently interact within TADs to regulate gene expression.

We further explored the role of CREs in the regulation of gene expression during hypoxic acclimatization. We first performed differential expression analysis between pairwise comparisons of low-altitude Hu sheep, high-altitude Hu sheep after translocation to the QTP for 8 months, and Tibetan sheep based on the RNA-Seq data. For each comparison, we annotated target genes linked to the DARs and detected the common genes showing both up- or down-regulated expression and changes in chromatin accessibility (Fig. 5e and Supplementary Data 29). We identified a total of 19,151 common peak-gene pairs between groups (i.e., low-altitude Hu sheep vs. high-altitude Hu sheep, low-altitude Hu sheep vs. Tibetan sheep and high-altitude Hu sheep vs. Tibetan sheep) across tissues, including 364 up-regulated and 18,787 down-regulated genes (Fig. 5f). We found that the common genes identified from the comparison of low-altitude Hu sheep vs. high-altitude Hu sheep were related to hypoxia adaptation. For example, down-regulated common genes in high-altitude Hu sheep were significantly (FDR < 0.05) enriched in the response to hypoxia and regulation of hemopoiesis in the liver (Fig. 5g and Supplementary Data 30). In particular, *PPARG*, whose expression was down-regulated due to less accessible chromatin (Fig. 5h), is relevant to the regulation of cardiovascular circadian rhythms[52]. Additionally, *LIMD1*, whose functions are associated with the regulation of hippo signaling[53] and the response to hypoxia[54], was up-regulated in the hypothalamus in high-altitude Hu sheep due to open accessibility (Fig. 5h). Therefore, the expression of the aforementioned common genes was regulated (i.e., up- or down-regulated) by chromatin accessibility and further affected hypoxia acclimatization.

## Acclimatization to high altitude in offspring

To explore the acclimatization to high altitude in offspring, we examined the values of SpO$_2$, gene expression, and chromatin accessibility in lambs and ewes of the three sheep groups (i.e., low-altitude Hu, high-altitude Hu, and Tibetan sheep). Strikingly, the high-altitude Hu lambs showed no significant difference in the mean value of SpO$_2$ measures from Tibetan lambs (high-altitude Hu lamb: 85.92, 95% confidence intervals: 83.81 − 88.03; and Tibetan lamb: 87.7, 95% confidence intervals: 85.87 − 89.55; Wilcoxon rank sum test, $P = 0.16$) (Fig. 6a). We performed differential expression analysis between pairwise comparisons of the three lamb groups and focused on the DEGs from the comparisons of high-altitude Hu lambs vs. low-altitude Hu lambs and Tibetan lambs vs. low-altitude Hu lambs (Fig. 6b). Based on comparison with low-altitude Hu lambs, we found that the DEGs detected in high-altitude Hu lambs and Tibetan lambs were significantly (FDR < 0.05) enriched in many common GO terms, such as extracellular matrix organization in kidney, localization within membranes in cerebrum and the cellular response to angiotensin in artery (Fig. 6c and Supplementary Data 31). Some of these GO terms (e.g., extracellular matrix organization and localization within membrane) were directly activated by hypoxia[55], suggesting that high-altitude Hu lambs and Tibetan lambs may share similar hypoxia-responsive

biological processes. Additionally, we noticed that several hypoxia response processes were only identified in the comparison of high-altitude Hu lambs vs. low-altitude Hu lambs, including the response to hypoxia in the lung and response to decreased oxygen levels in the artery and lung (Fig. 6d).

We further examined the expression profiles of the important functional genes *CAT* and *UCP3* in response to hypoxia in lung, *NPPC* in response to decreased oxygen levels and *HBB* in blood circulation in artery. The expression patterns of the genes were similar to the patterns of SpO$_2$ alterations (Fig. 6e). Moreover, we identified common genes showing both up- or down-regulated expression and changes in chromatin accessibility between the lamb groups for each sampled tissue (i.e., lung, heart and hypothalamus) (Supplementary Data 29). For example, the expression of functional genes for high-altitude adaptation, such as *SIK1, OTOF, SOCS1* and *JUN* in heart and *CXCL8*[56−58] in lung, showed significant ($P < 0.05$) downregulation in high-altitude Hu lambs compared to low-altitude Hu lambs. Overall, the results indicated that high-altitude Hu lambs show developed adaptive characteristics at birth according to the three above measures, presenting similar values to Tibetan lambs but significant differences from those of low-altitude Hu lambs. The hypoxia exposure of parents could account for the improved oxygen regulatory ability in their offspring under hypoxia stress.

## Expression of human high-altitude adaptive and disease genes

We first examined the similarity of global expression patterns between sheep and human. We retrieved publicly available RNA-Seq data from the human Genotype-Tissue Expression (GTEx) consortium (v8) and conducted comparative analysis using 17,279 one-to-one orthologous genes in 14 common tissues (i.e., hypothalamus, pituitary, cerebellum, ileum, colon, leukocyte, spleen, heart, muscle, artery, adipose, lung, liver and kidney). The *t*-SNE-based expression clustering among samples clearly recapitulated tissues rather than species (Fig. 7a). Similar results were observed in the hierarchical clustering of tissues based on median gene expression (Fig. 7b).

We collected candidate genes associated with high-altitude adaptation (e.g., Tibetan, Andean and Ethiopians) (Supplementary Data 32) and mountain sickness (e.g., pulmonary hypertension, polycythemia and heart failure) in human (Supplementary Data 33). We calculated the expression correlations of the adaptive and disease-related genes between sheep and human and found that most adaptive genes (97.82%, 1160 out of 1207 genes) and disease-related genes (97.43%, 580 out of 613 genes) were significantly correlated (Pearson's correlation, $P < 0.05$) (Supplementary Data 34). Therefore, we used the sheep time-series transcriptomic data to investigate the expression changes in these genes with time across tissues (Supplementary Figs. 19 and 20). For example, *BMPR2* is a well-known gene associated with pulmonary hypertension[59,60], and its expression levels were highly correlated (Pearson's $r = 0.96$, $P = 7.21 \times 10^{-8}$) between human and sheep across tissues (Fig. 7c). The expression changes in *BMPR2* in critical tissues such as lung, artery, and heart showed distinct patterns over time. Specifically, the expression level changed greatly at 7 d in lung and at 21 d in artery and gradually increased with time in the heart (Fig. 7c). Additionally, the expression levels of the high-altitude adaptive gene *HIF1A* displayed a high correlation (Pearson's $r = 0.86$, $P = 8.45 \times 10^{-5}$) between human and sheep (Fig. 7d). The expression of this gene fluctuated with time in the lung but gradually increased with time in heart and cerebellum (Fig. 7d). We noted that genes responsible for similar traits or diseases showed similar time-series expression patterns in relevant tissues (Supplementary Data 35). For instance, *APOLD1* and *KCNA5*, which are associated with pulmonary hypertension, showed similar expression patterns to *BMPR2* in lung, artery and heart (Supplementary Data 35). The adaptive genes *EPAS1* and *VEGFA* exhibited similar expression patterns to *HIF1A* in the cerebellum and kidney (Supplementary Fig. 20 and Supplementary Data 35).

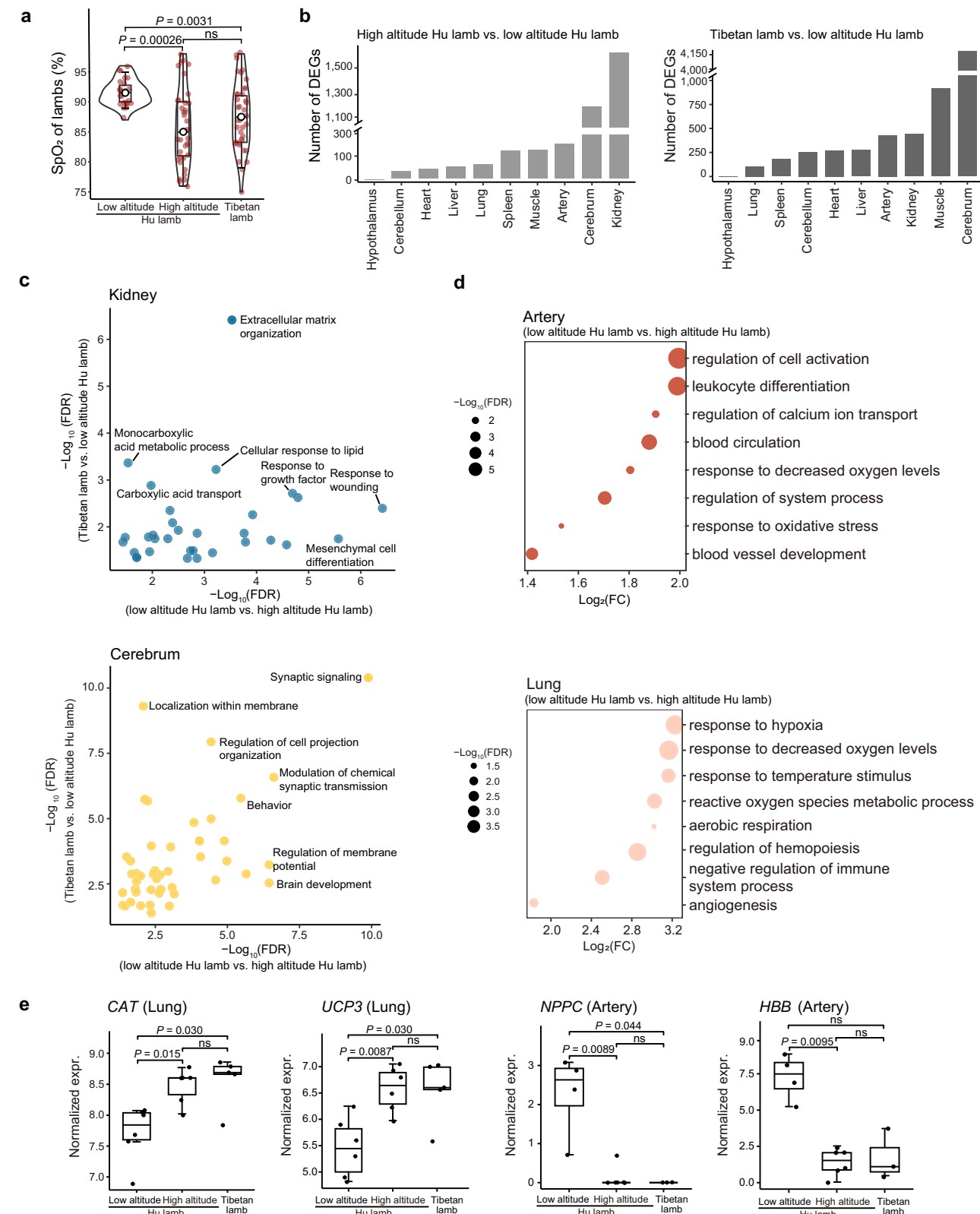

As the HIF and PHD genes are key regulators of high altitude adaptation, we further examined the expression patterns of three HIF genes (i.e., *EPAS1*, *ARNT* and *HIF1A*) and a PHD gene (i.e., *EGLN2*) in our sheep model. We found consistently and significantly (Wilcoxon rank sum test, $P < 0.05$) up-regulated expression of HIF genes *EPAS1*, *ARNT* and *HIF1A* and down-regulated expression of PHD gene *EGLN2* in the cerebellum in high-altitude Hu sheep (i.e., 7 d, 14 d, 21 d and 8 mon) as

compared to low-altitude Hu sheep (Supplementary Fig. 21). The expression of *EPAS1*, *EGLN2* and *ARNT* across tissues in low-altitude Hu sheep was significantly ($P < 0.05$) correlated with that in high-altitude Hu sheep, and *EPAS1* showed the highest correlation coefficient values (Pearson's $r > 0.97$) (Supplementary Fig. 22). For individual tissues, the expression of HIF genes *EPAS1*, *ARNT* and *HIF1A* in cerebellum showed the maximum extent of expression changes across the four

**Fig. 6 | Acclimatization to high altitude in offspring. a** Changes in $SpO_2$ in the three lamb groups. The average values with 95% confidence interval (CI) of $SpO_2$ in lamb groups are: low-altitude Hu lamb ($n = 6$), 91.50 (CI: 90.63 − 92.37); high-altitude Hu lamb ($n = 6$), 85.92 (CI: 83.81 − 88.03); and Tibetan lamb ($n = 6$), 97.71 (CI: 85.87 − 89.55). **b** Numbers of DEGs from the high-altitude Hu lamb vs. low-altitude Hu lamb (left) and Tibetan lamb vs. low-altitude Hu lamb comparisons (right). **c** Common GO terms enriched with the DEGs of low-altitude Hu lambs vs. high-altitude Hu lambs and low-altitude Hu lambs vs. Tibetan lambs in the kidney (top) and cerebrum (bottom). **d** GO terms enriched only with the DEGs from the low-altitude Hu lamb and high-altitude Hu lamb comparison in artery (top) and lung (bottom). **e** Expression levels of key genes from hypoxia response-related GO terms ($n = 6$). expr, expression. Boxplots are represented by minima, 25% quantile, median, 75% quantile, and maxima. Each dot represents individual expression level in different groups. $P$ values in the figures a and e come from the two-sided Wilcoxon rank sum test, "ns" indicates not significant. Source Data are provided as Source Data file.

acclimatization time points when low-altitude Hu sheep were translocated to high altitude (Supplementary Fig. 21a−c). We also observed that the expression levels of HIF genes *EPAS1*, *ARNT* and *HIF1A* in cerebellum were significantly (Wilcoxon rank sum test, $P < 0.05$) higher in Tibetan sheep than those in low-altitude Hu sheep (Supplementary Fig. 23). These results indicate that the expression patterns of HIF genes in cerebellum may play an important role in the acclimatization and adaptation to hypoxia.

Furthermore, we used lung scRNA-Seq data to dissect the particular cell types involved in high-altitude adaptation and diseases (Supplementary Data 36 and 37). We examined the expression of *HIF1A* and *BMPR2* across all cell types in lung tissues. We found high expression levels of *HIF1A* in classical monocytes (CMs) and club cells (CLU) and *BMPR2* in vein endothelial cells (VECs) (Fig. 7e). Additionally, we found that the time-series expression patterns of *HIF1A* and *BMPR2* were similar to those obtained from bulk RNA in lung (Fig. 7f). Cell-cell communication analysis showed that cell communication events continuously increased with time, in which proliferating T cells (PTCs), CMs and CLUs maintained high levels of communication. Moreover, we predicted that core transcription factors (TFs), such as NFKB1, RELB and CEBPB, regulate *HIF1A* and *BMPR2* (Fig. 7h). Notably, NFKB1, a transcription regulator of *HIF1A*, can be activated by oxidant-free radicals and ultraviolet irradiation[61], which is functionally relevant to high-altitude adaptation. The function of the *BMPR2*-related transcription regulator RELB is associated with the NF-kappa-B pathway, which is involved in disease-related processes such as inflammation, immunity, and tumorigenesis[62].

## Discussion

Using sheep as a model, we have created the first comprehensive time-series transcriptome atlas of whole-body major tissues for lowland animals translocated to a high-altitude environment and transcriptomes for their offspring born at high altitudes. Leveraging these critical data, we are able to explore the gene expression patterns of major tissues during the acclimatization process, which offers an exceptional opportunity to dissect the differences in the regulatory mechanisms underlying genetic adaptation and short-term acclimatization to hypoxia as well as the transcriptional changes responsible for the acclimatization in offspring. The sheep time-series transcriptomes also provide valuable resources for depicting the temporal expression of genes associated with human high-altitude adaptation and diseases.

The high-altitude hypoxia adaptation of indigenous highland inhabitants has always been related to the morphological and functional remodeling of the lungs, heart, and artery[3,63,64]. However, in the short-term acclimatization of high-altitude Hu sheep translocated from lowlands, we found that the cerebellum, kidney, adipose, muscle, colon, ileum, blood and spleen were among the most active tissues in response to hypoxia stress based on the numbers of DEGs identified between adjacent time points (Fig. 3a) and DCGs across the examined time points (Fig. 3h). This provided clear evidence that different body systems involving distinct tissues and consequently different strategies for oxygen utilization should be required for long-term adaptation and short-term acclimatization to hypoxia, respectively. For instance, the nervous (cerebellum), metabolic (kidney, adipose, muscle), digestive (colon, ileum), and immune (blood, spleen) systems seem to make major contributions to short-term hypoxia acclimatization

through the coregulation of central coordination, energy metabolism, intestinal homeostasis, and immune response (Fig. 3i and Supplementary Data 17). These systems and tissues may have to reduce oxygen consumption under hypoxia because they are oxygen-consuming parts of the body[65–67]. In contrast, the respiratory (lung) and cardiovascular (heart and artery) systems are mainly responsible for long-term hypoxia adaptation, and they can improve the exchange and transportation of oxygen in response to hypoxia[68,69]. Thus, multiple systems and tissues actively respond to short-term hypoxia acclimatization because short exposure to hypoxia can stimulate the stress response of the whole body, leading to dramatic physical adjustments[34,70]. By integrating the expression profiles and bioindicators of blood, we identified 10 gene modules (e.g., M3) that were significantly correlated with informative bioindicators of the hypoxia response, such as erythropoietin (EPO), nitric oxide synthetase (NOS), glutathione peroxidase (GSH-Px) and alkaline phosphatase (ALP) (Fig. 2e). The functions of these bioindicators suggested that the transcriptional regulation of oxygen transportation (EPO, NOS)[71,72], antioxidation (GSH-Px)[73] and energy metabolism (ALP)[74] through the blood and circulatory systems may also contribute to short-term hypoxia acclimatization. Notably, the level of $SpO_2$, one of the most important indicators of blood oxygen content, recovered with acclimatization across time points (Fig. 2a), indicating the successful gradual acclimatization of high-altitude Hu sheep through the collaborative regulation of the aforementioned different systems and tissues.

From the whole-genome selection test between low-altitude and high-altitude sheep together with differential expression analysis across tissues, we identified 52 $F_{ST}$ genes that were differentially expressed between Tibetan sheep and low-altitude Hu sheep, and 179 $F_{ST}$ genes that showed dynamic changes in expression during the acclimatization of translocated Hu sheep in at least five tissues (i.e., multi-tissue $F_{ST}$ gene) (Fig. 4a-c). These two panels of multi-tissue $F_{ST}$ genes appeared to be putatively selected in the genome and effectively expressed in multiple tissues, providing new clues for delineating the genetic basis of long-term adaptation and short-term acclimatization to high-altitude hypoxia at the multi-tissue level. Notably, we found 13 genes (e.g., *APOLD1*, *NDUFB9*, *ERBB4*, *NFKBIZ*, *NR4A3*, *RPS8*, *CIAO2A*, *AHCYL2*, *ESRRG*, *KIAA0930*, *RASGEF1B*, *MRPS25* and *TNFRSF21*) that were shared between the two panels (Fig. 4c), which could represent reliable candidates for both hypoxia adaptation and acclimatization at the genomic and transcriptional levels. For example, *APOLD1* and *NR4A3* showed significant expression changes in many tissues (Fig. 4a, b), and they were reported to be functionally associated with cardiopulmonary, metabolic, and neurological phenotypes[75–78] (Supplementary Data 25). This implied roles of these genes in hypoxia adaptation (cardiopulmonary changes) and acclimatization (metabolic and neurological adjustments). Moreover, we found that the overlapped genes between long-term adaptation and short-term acclimatization are significantly higher than the random overlaps (Supplementary Fig. 15). Previous studies usually investigated short-term and long-term responses to environmental stress separately, but the interaction between the two processes has been rarely evaluated, e.g., whether short-term transcriptional response may facilitate or constrain long-term genetic adaptation[11,79]. Our results provided a valuable example that short-term acclimatization could facilitate

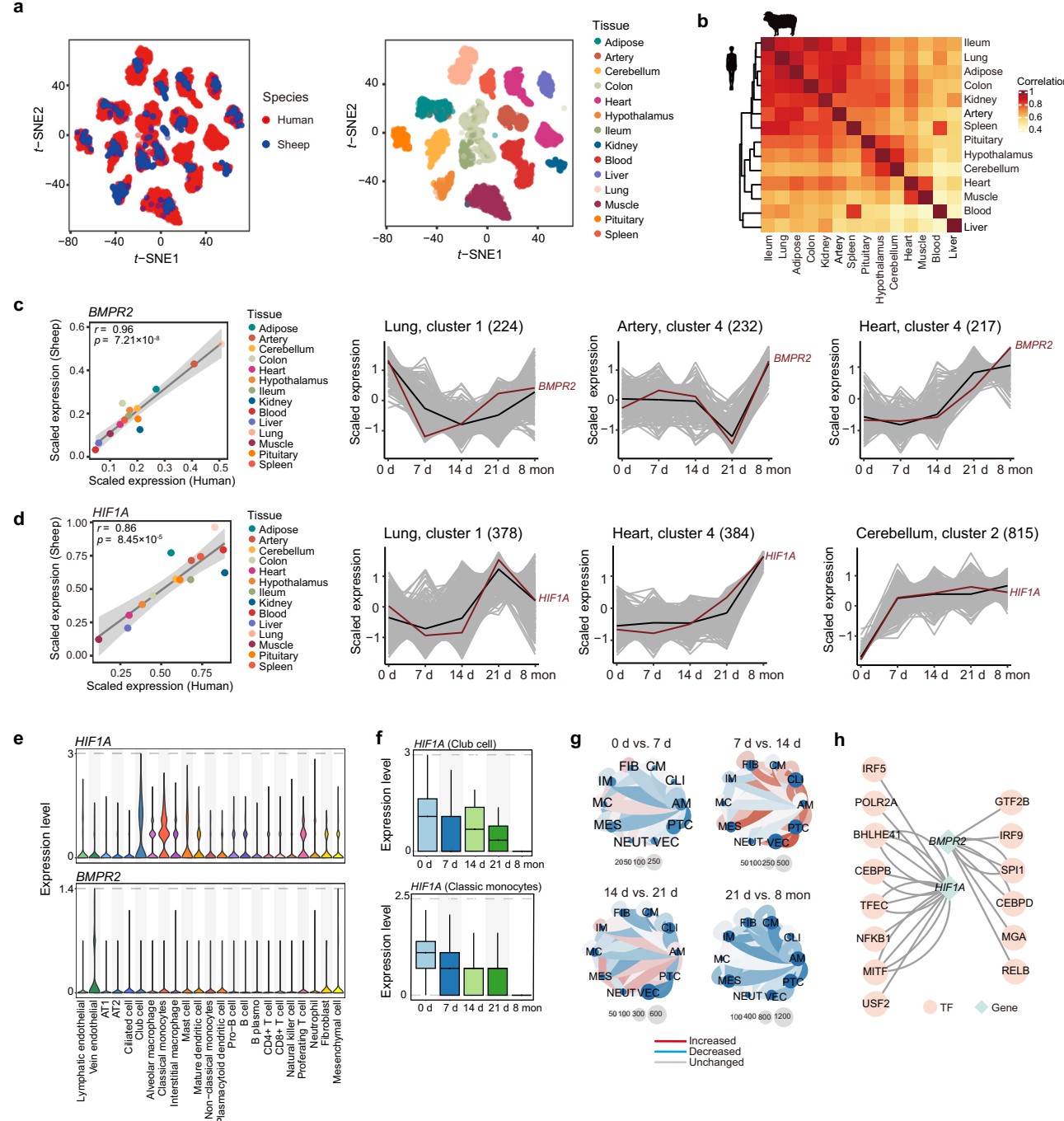

**Fig. 7 | Time-series transcriptome of genes implicated in adaptation and disease in human. a** Conservation of transcripts of 14 common tissues in human and sheep. *t*-SNE clustering of samples in our study (*n* = 1277) and the human GTEx v8 consortium (*n* = 6792) based on batch-corrected expression. Species (left) and tissue types (right) are distinguished by color. **b** Hierarchical clustering of common tissues in humans and sheep based on Pearson's correlation of the median TPM value. **c, d** Gene examples for human pulmonary hypertension (i.e., *BMPR2*) and high-altitude adaptation (i.e., *HIF1A*). **c** Pearson's correlation between humans and sheep based on the median value of *BMPR2* (left). Expression patterns of *BMPR2* in crucial tissues over time (right). The two-sided *P* values are calculated by the linear regression model. Shading: standard error of the fitting line. d, Similar to c, but for

the *HIF1A* gene. **e** The expression of *BMPR2* and *HIF1A* across cell types in the lung. AT1, alveolar type 1 cell; AT2, alveolar type 2 cell. **f** The expression of HIF1A in club cells (top) and classic monocytes (bottom) over time in the lung with a single replicate (*n* = 1). Boxplots are represented by minima, 25% quantile, median, 75% quantile, and maxima. **g** Cell-cell communication results for differentially expressed cell types from *BMPR2* and *HIF1A* in adjacent time point comparisons. FIB, fibroblast; CM, mast cell; CLI, ciliated cell; AM, alveolar macrophage; PTC, proferating T cell; VEC, vein endothelial; NEUT, neutrophil; MES, mesenchymal cell; MC, mast cell; IM, interstitial macrophage. **h** Transcription factors (TFs) regulating *HIF1A* and *BMPR2*. Source Data are provided as Source Data file.

long-term adaptation to high-altitude hypoxia. Additionally, we explored the roles of CREs in gene expression by integrating differential expression and differential chromatin accessibility analysis. Among the common genes showing both up- or down-regulated expression and changes in chromatin accessibility, most genes in high-

altitude Hu sheep and Tibetan sheep were down-regulated compared with those in low-altitude Hu sheep across tissues (Fig. 5f and Supplementary Data 29).

Oxygen sensing machinery can be subject to both positive (e.g., up-regulation) and negative regulatory (e.g., down-regulation)

feedbacks[80]. Our results were consistent with previous findings showing that negative feedback loops, e.g., down-regulation of angiogenesis-associated genes (e.g., *PPARG*) in deer mice and down-regulation of high-altitude adaptation genes (e.g., *EPAS1*) in Tibetans, play an important role under hypoxic stress[81,82]. Additionally, previous studies investigated hypoxia-induced changes to chromatin accessible regions and associated gene expressions, and revealed the impact of chromatin landscape on gene regulation across different cell types (e.g., HeLa cells, HL-1 cells and HUVEC cells)[82–84]. Here, we evaluated the chromatin accessibility in low-altitude Hu sheep and its correlation with the expression of differentially expressed genes in translocated high-altitude Hu sheep across tissues. The low to moderate correlation coefficients ($r = 0.33 - 0.49$, $P > 0.05$) from the linear regression (Supplementary Fig. 24) implied that differences in OARs across tissues in low-altitude Hu sheep did not have a significant impact on the expression of DEGs across tissues in translocated high-altitude Hu sheep. With regard to the chromatin and expression changes within high-altitude Hu sheep, we provided an exceptional example of whole-genome chromatin dynamics in response to hypoxia, with chromatin accessibility mostly decreased by down-regulation, which subsequently repressed the expression of most genes. The above findings could contribute to our understanding of the molecular mechanisms dictating gene repression in hypoxia at the transcriptomic and epigenomic level.

Interestingly, we found that high-altitude Hu lambs exhibited $SpO_2$ values and gene expression patterns similar to those of Tibetan lambs (Fig. 6a, e), implying that high-altitude Hu lambs had already acclimatized to high-altitude hypoxia at birth. Notably, the DEGs detected in the comparisons of high-altitude Hu lambs vs. low-altitude Hu lambs and Tibetan lambs vs. low-altitude Hu lambs were enriched in some common GO terms (e.g., extracellular matrix organization in the kidney, localization within membrane in the cerebrum and cellular response to angiotensin in artery) that could be directly activated by hypoxia[55] (Fig. 6c and Supplementary Data 31). The observation suggested similar genetic regulation of relevant tissues (e.g., cerebrum, kidney, and artery) in response to hypoxia in high-altitude Hu lambs and Tibetan lambs. The enriched functions of DEGs also showed GO categories specific to high-altitude Hu lambs in response to hypoxia in the lungs and artery (Fig. 6d and Supplementary Data 31). This may indicate that high-altitude Hu lambs descended from short-term hypoxia-acclimatized parents should require more transcriptional regulation and physical adjustments to cope with hypoxia than indigenous Tibetan lambs. It should be noted that at the current time, we can't distinguish whether the successful acclimatization of Hu lamb was due to direct antenatal exposure to low oxygen in utero or a true heritable effect. Compared with humans[85] and other lowland animals translocated to high altitudes (e.g., cattle[86] and mouse[87]), sheep showed much less fetal growth restriction and neonatal mortality based on both the observation made in our translocation experiment (neonatal mortality: 26.32%) and previous reports[88]. Since hypoxia may increase stillbirth and infant mortality[89,90], the DEGs identified in high-altitude Hu lambs may hold the potential to dissect the genetic basis of low mortality of offspring under hypoxia and thus contribute to the improvement of pregnancy outcomes in human[22] and other animals.

Our comprehensive transcriptome data for major tissues in the sheep model showed high correlations with human GTEx data according to tissue type, based on the expression clustering of 17,279 one-to-one orthologous genes in 14 common tissues (Fig. 7a, b). This demonstrated that our sheep expression atlas could be used to improve the interpretation of the genetic mechanisms underlying hypoxia-related adaptation and diseases in humans. At the gene level, the expression levels of two representative human genes (i.e., *BMPR2*, associated with pulmonary hypertension[59,60], and *HIF1A*, changes in gene expression are associated with high-altitude hypoxia adaptation[91,92] exhibited significantly high correlations in relevant

tissues between human and sheep (Fig. 7c, d and Supplementary Data 34), further demonstrating the rationality of employing sheep expression profiles for illustrating the expression patterns of human adaptation and disease genes. Importantly, the time-series characteristics of sheep transcriptomes may be valuable for providing missing information about temporal expression changes in the aforementioned human genes (Fig. 7c, d). We expect that our time-series sheep transcriptomes reflect analogous dynamic expression changes in investigated human genes at the tissue and cell levels (e.g., lung).

Apart from the findings stated above, it is worth discussing potential limitations to the present study. To warrant the credibility of the whole-genome selection test, we used multiple analyzes to demonstrate that Hu sheep and Tibetan sheep used in this study are pure breeds. Our Hu sheep were clustered with the pure individuals from low altitude from phylogenetic tree analysis (Supplementary Fig. 25) and exhibited the closest genetic distance with Tibetan sheep (Supplementary Fig. 26) and Supplementary Data 38). For Tibetan sheep, we didn't detect any interbreed introgression from 16 Chinese sheep breeds (Supplementary Fig. 27) or interspecific introgression from sympatric wild sheep (e.g., argali)[93] based on the Structure analyzes (Supplementary Fig. 28), ABBA-BABA (Supplementary Fig. 29) and gene tree (Supplementary Fig. 30). Following these findings, we excluded the potential impact of phylogenetic relationship or genetic introgression on the whole-genome selection test between high-altitude Tibetan sheep and low-altitude sheep (e.g., Hu sheep) in our study. Nevertheless, we cannot completely rule out the possibility of introgression and its effect on high altitude adaptation of Tibetan sheep. First, although introgression was not detected in Tibetan sheep used in this study, such introgression may be identified in future studies when more individuals of Tibetan sheep are included and analyzed. Second, our introgression analyzes only utilized SNPs and did not consider other types of genetic variations (e.g., SVs). A recent study found that the introgressed segments between yak and cattle related to altitude adaptation were only revealed based on SVs in a few key genes (e.g., *EPAS1*)[10]. SVs could contribute to the high altitude adaptation of both animals (e.g., yak)[10] and plants (e.g., *Arabidopsis thaliana*)[94]. For instance, an insertion in the promoter region of the *HPCA1* gene enhances gene expression and promotes the adaptation of *Arabidopsis thaliana* to alpine environments[94]. Despite the importance of SVs, it is unfortunate that we don't have long-read sequencing genomic data and a large panel of samples to yield precise and comprehensive SVs for corresponding analysis in this study. Also, the focus of the present study is to reveal transcriptional regulation in response to high-altitude hypoxia, therefore genomic analysis on SVs is beyond our main topic. Future studies focusing on SVs may discover whole-genome SV characteristics and the impact of SVs on introgression, gene expression, and high-altitude adaptation in sheep.

In conclusion, we generated time-series transcriptome resources of major tissues in a sheep model for high-altitude- or hypoxia-related studies. We identified a credible set of active tissues and crucial genes for the short-term hypoxia acclimatization of sheep and for the acclimatization of their offspring. These tissues and genes likely function in multiple body systems and may work together to shape adaptive or maladaptive traits in response to hypoxia. We further utilized sheep time-series transcriptomes to mirror the dynamic expression changes in high-altitude adaptation and disease genes of humans, which will probably provide insights into the molecular mechanisms underlying human adaptation and diseases. Our study demonstrates how multi-tissue expression profiles across time can be used to inform multiple aspects of short-term acclimatization and disease interpretation.

## Methods
### Ethical statement
All experimental protocols in this study were reviewed and approved by the Institutional Animal Care and Use Committee of China

Agricultural University (CAU20160628-2) and the local animal research ethics committee. Animal care, maintenance, procedures, and experimentation were performed in strict accordance with the guidelines and regulations approved by the Welfare and Ethics Committee of the Chinese Association for Laboratory Animal Sciences.

## Experimental design

Individuals of Hu sheep and Tibetan sheep, two representative Chinese native breeds that originally inhabited low-altitude regions (Zhejiang Province, China) and the high-altitude regions [the Qinghai-Tibet Plateau (QTP), China], respectively, were included in the experiment. Overall, adult ewes (~1.5 years) and lambs (~2 months) of the two breeds with good health and body condition were raised under three different scenarios: native in a low altitude region (scenario 1), translocated from a low altitude to high altitude (scenario 2), and native on a high altitude (scenario 3) (Fig. 1a). In scenario 1, 10 adult ewes (~1.5 years) and 6 lambs (~2 months) of Hu sheep were housed at ~350 m.a.s.l. on the Wanghu Livestock Farm in Neijiang City, Sichuan Province, China. In scenario 2, 40 adult ewes and 3 rams of Hu sheep born and raised in the livestock farm under scenario 1 were translocated to the Tibetan Sheep Breeding Farm of Sichuan Province (Aba Tibetan and Qiang Autonomous Prefecture, Sichuan Province, China) and produced 10 lambs after approximately 8 months. In scenario 3, 10 adult ewes and 6 lambs of Tibetan sheep were housed at ~3500 m.a.s.l. on the sheep farm under scenario 2. Tibetan sheep have inhabited the QTP for approximately 4000 years[29,95]. All animals raised in experimental farms were under standard and uniform housing and feeding conditions. Animals were fed twice per day with formula diets containing 16% crude protein, 5% crude fiber, 7% crush ash, 0.6% calcium, 0.7% lysine and 0.5% phosphorus and had *ad libitum* access to water and mineral salt. The details of the experimental animals are summarized in the Supplementary Data 2.

## Collection of blood biochemical data and animal tissues

Phenotypic data of 20 blood parameters (i.e., blood oxygen saturation, glucose, triglycerides, alanine transaminase, aspartate aminotransferase, total bilirubin, alkaline phosphatase, lactate dehydrogenase, uric acid, blood urea nitrogen, creatinine, cardiac enzymes, calcium, superoxide dismutase, glutathione peroxidase, catalase, nitric oxide, malondialdehyde, erythropoietin and nitric oxide synthase) and samples from 19 whole-body tissues [i.e., heart, liver, spleen, lung, kidney, rumen, abomasum, duodenum, ileum, jejunum, colon, cerebrum, cerebellum, hypothalamus, pituitary, artery, muscle, adipose and blood (leukocyte)] were collected from all the animals in the above three scenarios. In scenario 2, data and tissues of the 40 Hu sheep ewes were collected at four sequential time points [7 days, 14 days, 21 days and ~8 months (245 days)] after their translocation to the QTP, with 10 ewes sampled at each time point. In particular, 10 tissues of the 6 lambs of Hu sheep born on the QTP were collected at an age of ~2 months, approximately 8 months after transportation (Fig. 1a). In scenarios 1 and 3, the same phenotypic data and tissues of 10 ewes and 6 lambs of Hu sheep (scenario 1) and 10 ewes and 6 lambs of Tibetan sheep (scenario 3) were collected (Supplementary Data 7). Blood oxygen saturation (SpO$_2$) values and 19 additional blood biochemical indicators were examined in the animals (Supplementary Data 8). Arterial SpO$_2$ was measured using the Tough/Ear Blood Oxygen Metre Veterinary SpO$_2$ PR Monitor (RocSea, Jingzhou, China) when the animals were stationary. Blood samples were collected with a 5 mL vacuum tube and were centrifuged immediately to isolate plasma. The isolated plasma samples were then stored in a −80 °C freezer and used to measure the 19 blood biochemical indicators with commercial assay kits (Jincheng Bioengineering Inc., Nanjing, China).

Animals were slaughtered by carotid artery exsanguination. Following sacrifice, tissues were isolated and placed on an ice board for dissection. Each tissue was cut into 5 – 10 pieces of approximately

50 – 200 mg each. Samples were then transferred into 2 mL cryogenic vials (Corning, NY, USA, Cat. No. 430917), snap frozen in liquid nitrogen, and stored until DNA extraction for WGS or RNA extraction for RNA-Seq. In total, 49 samples from heart tissue of 49 sheep (10 ewes of Hu sheep in the low altitude region and 39 ewes of Hu sheep translocated to the QTP) were collected for whole genome sequencing (Fig. 1b). A total of 1277 samples from 19 various tissues of 78 sheep were collected for bulk RNA-Seq (Fig. 1b). Additionally, 66 samples from eight tissues (i.e., hypothalamus, rumen, heart, lung, liver, duodenum, spleen and adipose) of 12 sheep were collected for ATAC-Seq, including 18 samples of three tissues (lung, heart and hypothalamus) from six lambs (two lambs of Hu sheep in the low altitude region, two lambs of Hu sheep in the QTP, and two lambs of Tibetan sheep in the QTP) and 48 samples of eight tissues from six ewes (two ewes of Hu sheep in the low altitude region, two ewes of Hu sheep in the QTP, and two ewes of Tibetan sheep in the QTP) (Fig. 1b).

Six samples (i.e., low-altitude Hu sheep in scenario 1, high-altitude Hu sheep at four acclimatization time points in scenario 2, and Tibetan sheep in scenario 3) from different parts of the lung (left lung and right lung) were harvested and then cleaned with PBS for scRNA-Seq. For each sample, sliced tissues were stored in tissue storage solution (Miltenyi Biotec, Bergisch Gladbach, Germany, Cat. No. 130-100-008) at 4 °C for single-cell suspension preparation and library construction.

## DNA extraction, library preparation and sequencing

DNA was extracted from flash-frozen heart tissue using the Tissue kit (QIAGEN, Shanghai, China). DNA integrity was evaluated on agarose gels and Qubit® DNA Assay Kit in Qubit® 3.0 Flurometer (Invitrogen, USA). Library construction and data sequencing were implemented by the Illumina platform. DNA libraries were constructed using TruSeq Library Construction Kit (Illumina, San Diego, USA). In brief, the DNA was fragmented, end polished, A-tailed, and ligated with the full-length adapter. The length of 350 bp fragments were selected, PCR amplified and purified with AMPure XP system (Beckman Coulter, Beverly, USA). The prepared libraries were examined insert size using Agilent 2100 Bioanalyzer (Agilent Technologies, CA, USA) and amplified. Then sequencing was implemented on the Illumina Novaseq 6000 platform by Novogene Co., Ltd. (TianJin, China). 150 bp paired-end reads with a target depth of ~20-fold coverage per genome were generated.

## RNA extraction, library preparation and sequencing

Total RNA was extracted from flash-frozen tissues with RNA TRIzol (Invitrogen, Carlsbad, CA, USA) according to the manufacturer's protocol. After purification, RNA quality was checked using agarose gel electrophoresis and a NanoPhotometer® spectrophotometer (IMPLEN, CA, USA). RNA integrity (RIN) was examined on an Agilent 2100 Bioanalyzer (Agilent Technologies, Waldbronn, Germany) with a cut-off of an RIN < 7.00, and the RNA concentration was measured with the Agilent 2100 RNA 6000 Nano Kit (Agilent Technologies, Waldbronn, Germany). First-strand cDNA was generated using the FastKing One-Step RT–PCR Kit (TIANGEN Biotech, Beijing, China), and cDNA libraries were constructed by the Illumina TruSeq RNA Library Prep Kit v2 (Illumina, CA, USA). RNA-Seq was implemented on the Illumina HiSeq 2500 (Illumina, CA, USA) at Novogene Co., Ltd. (TianJin, China), generating 150 bp paired-end reads.

## ATAC library construction and sequencing

Library preparation for ATAC-Seq followed a modified OmniATAC protocol[96] in cryopreserved nuclei. Specifically, weighed frozen tissues (~20 mg) were first lysed in cold homogenization buffer (10 mM Tris-HCl, pH 7.4, 10 mM NaCl, 3 mM MgCl$_2$, 0.1% Igepal). Nuclei were then resuspended and collected from the interface after iodixanol-based density gradient centrifugation. Thereafter, nucleus tagmentation was performed in Tn5 transposase reaction mix (Illumina Tagment DNA Enzyme and Buffer kits) under incubation at 37 °C for 30 min in a

thermomixer with shaking at 1000 rpm, and two equimolar adapters were added. Immediately following the transposition reactions, DNA was purified with the Qiagen MinElute PCR Purification Kit (Qiagen, Netherlands, Cat. No. 28004) and eluted in EB buffer. To amplify the library, PCR was then performed in a mix of 10 μM Nextera i7 and i5 primers and NEBNext Q5 High-Fidelity PCR Master Mix (New England Biolabs, MA, USA) according to the following protocol: 72 °C for 5 min, 98 °C for 30 sec, and 11 cycles of 98 °C for 10 sec, 63 °C for 30 sec and 72 °C for 1 min. PCR products were purified with the Qiagen MinElute PCR Purification Kit and AMPure XP beads (Beckman Coulter, Cat. No. A63880) and resuspended in ultrapure nuclease-free distilled water. Library quality was assessed with a Qubit 2.0 system (Life Technologies, MA, USA), and fragment size was examined using an Agilent 2100 Bioanalyzer. The libraries were sequenced on an Illumina NovaSeq 6000 system with a 150 bp paired-end sequencing method.

### scRNA-Seq library construction and sequencing
scRNA-Seq libraries of lung tissues were constructed following previous protocols with minor modifications[97–99]. For the lung tissues, gentle and rapid generation of single-cell suspensions was achieved by using a modified version of the procedure of a mouse Lung Dissociation Kit (Miltenyi Biotec, Bergisch Gladbach, Germany; Cat. No. 130-095-927). In summary, we dissected sheep lung tissue into single lobes and rinsed the lobes in petri dishes containing PBS (pH = 7.2) to remove residual vessels, blood clots and mucin. Clean lobes were subjected to shacking digestion at 37 °C for 25 – 30 min with enzyme mix, which consisted of 2.4 mL of 1× Buffer S, 100 μL of Enzyme D, and 15 μL of Enzyme A. The digestion solution was briefly centrifuged at $600 \times g$ for 2 min at 4 °C, and the precipitated pellet was resuspended in 2.5 mL 1× Buffer and filtered with a 70 μM MACS SmartStrainer (Miltenyi Biotec, Bergisch Gladbach, Germany; Cat. No. 130-098–462). Then, the obtained cell suspension was centrifuged at $300 \times g$ for 10 min at 4 °C. After removing the supernatant, the cell pellet was resuspended in an appropriate buffer to the required volume for scRNA-Seq.

Qualified single-cell suspensions containing at least 8000 cells were loaded onto a chromium single-cell controller (10× Genomics), and single-cell gel beads were generated in the emulsion according to the manufacturer's protocol. Then, scRNA-Seq libraries were constructed using Single Cell 3' Library and Gel Bead Kit v3.1 (8000 initial cell capture number) and were subsequently sequenced using a NovaSeq 6000 sequencer (Illumina).

### Whole-genome sequence (WGS) data
**Data collection.** Whole-genome sequences were consisted of 49 WGS (average depth = ~15×) generated in this study and 152 WGS (average depth = ~17×) representing 16 Chinese native sheep breeds and one wild species retrieved from previous studies[29,30,100–102]. Detailed information on the populations, including the names, sampling locations and number of samples, was summarized in Supplementary Data 3, 9 and 39.

**Variant calling.** SNP calling followed previous protocols[30]. First, we filtered low-quality bases and artifact sequences using Trimmomatic (v.0.36)[103] and aligned the high-quality paired-end reads (150 bp or 100 bp) to the sheep reference genome *Oar_rambouillet_v1.0.* (GCA_002742125.1) using BWA (v0.7.8)[104] with the default parameters. Next, we removed duplicates in the BAM files using the *MarkDuplicates* module in GATK (v4.1.2.0)[105]. SNPs were then detected using the GATK *HaplotypeCaller* module with the GATK best-practice recommendations. Thereafter, we merged the GVCFs files called individually by the *CombineGVCFs* module and called SNPs with the *GenotypeGVCFs* module. Finally, we selected the raw SNPs using the *SelectVariants* module and filtered them using *VariantFiltering* of GATK with the parameters (QUAL < 30.0 || QD < 2.0 || MQ < 40.0 || FS > 60.0 || SOR > 3.0 || MQ RankSum < −12.5 || ReadPosRankSum < −20.0).

**SNP quality control.** SNP quality control was conducted with the following criteria using VCFtools (v0.1.17)[106]: 1) call rate > 90% and 2) minor allele frequency (MAF) > 0.05. Any SNPs that failed to meet any of the above criteria were filtered, and we obtained 833,880,778 SNPs for downstream analysis.

### Hi-C data preprocessing
**Quality control and data preprocessing.** Hi-C data from the blood of sheep were retrieved from the NCBI Sequence Read Archive (SRA) under accession number SRR19426890. We first trimmed adapter sequences and low-quality reads with Trimmomatic (v.0.36)[103] and obtained ~780 million clean reads. Hic-Pro (v2.9.0)[107] was then used to process the Hi-C data from raw sequencing data via a pipeline including alignment, matrix construction, matrix balancing, and iterative correction and eigenvector decomposition normalization (ICE) with the default parameters.

**Detection of TADs.** To explore the ATAC-Seq peak-to-gene linkage, we identified topologically associating domains (TADs) as follows. We first implemented the conversion of Hi-C matrices to Cooler format via HiCPeaks (v0.3.2)[108]. To detect the TADs, we then calculated the directionality index (DI) with a resolution of a Hi-C matrix at 40 kb using the *hitad* function from TADLib (v0.4.2)[109]. In total, we obtained 3032 TADs for subsequent analysis.

### RNA-Seq data preprocessing
Raw RNA-Seq reads with low base quality scores (quality scores ≤ 20) were first trimmed, and then adapter contamination was further removed using fastp (v0.20.1)[110]. The high-quality clean reads were next mapped to the sheep reference genome *Oar_rambouillet_v1.0* (GCA_002742125.1) by the program STAR (v2.7.9a)[111] with the settings (-quantMode GeneCounts, -outFilterMismatchNmax 3, -outFilterMultimapNmax 10). Properly paired and uniquely mapped reads were extracted by using SAMtools (v1.11)[112] with the command (view -f 2). Gene counts were generated by the featureCounts program from the Subread package suite (v2.0.3)[113]. We also normalized the raw counts of genes using the transcripts per million (TPM) method with an in-house script.

### ATAC-Seq data preprocessing
Raw ATAC-Seq reads were trimmed for Nextera adapters by using fastp (v0.20.0)[110] with the options (-q 15 -l 18), and the clean reads were aligned to the sheep reference genome *Oar_rambouillet_v1.0* using BWA (v0.7.17)[104] with the default parameters. PCR duplicates were removed using *Picard* module from GATK (v4.1.2.0)[105], and uniquely mapped high-quality reads were collected using SAMtools (v1.11)[112] with the options (view -f 2 -q 30). The reads mapped to the mitochondrial genome were also discarded, and the final BAM files were kept for subsequent analysis.

ATAC-Seq peaks were called by Genrich (v0.6.1)[114] with the options (-j -p 0.01 -b). Peak calling was first implemented in each library and then for each tissue based on concatenating all the replicates by using BEDTools (v2.30.0)[115] with the *bedtools merge* function. The *P* value was calculated for each peak assuming a null model with a log-normal distribution and corrected based on the Benjamini-Hochberg model.

The following parameters were measured as recommended by the ENCODE project (https://www.encodeproject.org/atac-seq/# standards) for the validation of ATAC-Seq libraries. The fraction of reads in peaks (FRiP) scores, nonredundant fraction (NRF) and other quality metrics (e.g., PCR bottlenecking coefficients, PBCs) for each sample were calculated with either SAMtools (v1.11)[112] or in-house scripts. Details of the quality control scores are included in Supplementary Data 5. Global correlations between samples were calculated with the R package DiffBind (v3.2.7)[116].

To generate consensus peaks, peaks for individual samples were merged with the *multiBigwigSummary BED-file* function in deepTools (v3.5.0)[117]. BAM files were converted into a normalized coverage track of bigWig format using the *bamCoverage* command in deepTools with the options (--binSize 10 --normailzeUsingRPKM -centerReads), and ATAC peaks were then visualized with Integrative Genomics Viewer (IGV) software (v2.9.4)[118]. Raw read counts in these peaks were determined by *multiBamSummary BED-file* function and were normalized for reads per kilobase per million (RPKM) reads with the function *normalize.quantiles* in the R package preprocessCore (v1.40.0)[119]. Additionally, *t*-SNE clustering of the ATAC-Seq profile was performed as described for the *t*-SNE analysis of the RNA-Seq data above.

## Population genetics analysis
We filtered the SNP data set with a MAF < 0.01, Hardy-Weinberg equilibrium < 0.001 and a proportion of missing genotypes > 0.05. We implemented LD pruning with the PLINK (v1.90)[120] option (indep-pairwise 50 5 0.05). After the filtering and linkage disequilibrium (LD) pruning, 8,445,599 SNPs were retained for population genetics analyzes. First, we calculated the identity-by-state (IBS) genetic distance matrix between the individuals using the PLINK (v1.90)[120] (−distance 1-ibs) and visualized the IBS distance matrix by an unrooted neighbor-joining (NJ) phylogenetic tree using the SplitsTree (v4.18.3)[121]. $F_{ST}$ distance between groups were estimated using *smartpca* function in EIGENSOFT (v.6.0.1)[122].

## Whole-genome selective sweep tests
To identify potential selective signatures associated with hypoxia adaptation between populations in high- and low-altitude regions, we selected domestic sheep from plateau ($n = 17$) and plain ($n = 20$) areas (Supplementary Data 18). We calculated genome-wide pairwise $F_{ST}$ values[123] between the high- and low-altitude populations with a sliding window approach (10 kb sliding windows with 10 kb steps) using the Python script *popgenWindows.py* (https://github.com/simonhmartin/genomics_general). Regions with the top 5% of the average $F_{ST}$ distribution were defined as selective signatures and then annotated to corresponding gained genes.

## Introgression analysis
We tested for the presence of interpopulation introgression from the other Chinese native sheep breeds into Tibetan sheep and interspecific genetic introgression from wild relatives into Tibetan sheep (Supplementary Data 39 and 40) using *D* statistics (i.e., ABBA-BABA statistics)[124]. We calculated the *D* statistics for the four-taxon model (H1, H2, H3, H4) using the function *qpDstat* implemented in the AdmixTools (v7.0.1)[125]. For the interpopulation introgression analysis, we used Menz sheep as the reference population (H1), Tibetan sheep as the target population (H2), individual populations of Chinese native sheep as the donor (H3), and ancestral alleles as outgroup (H4)[30]. For the interspecific introgression analysis, we used Menz sheep as the reference population (H1), Tibetan sheep as the target population (H2), Argali as the donor (H3), and ancestral alleles as outgroup (H4)[30]. The statistical significance of the *D* statistics was evaluated using a two-tailed Z-test, and |Z-score| > 3 was set as the threshold of statistical significance.

We also used the sNMF (v1.2)[126] to conduct Structure analysis for Tibetan sheep and argali. Additionally, we examined the possibility of interspecific introgression between argali and Tibetan sheep in the top selective regions between low-altitude sheep and Tibetan sheep. We extracted the SNPs located in the genomic regions encompassing the top 5 $F_{ST}$ gene (i.e., 50 kb up and downstream), and then constructed gene tree for each of these genes with SplitsTree (v4.18.3)[121].

## Tissue specificity of gene expression and chromatin accessibility
First, we clustered all 1277 samples based on TPM and the *t*-distributed stochastic neighbour embedding (*t*-SNE) method as implemented in

the R package Rstne (v0.16)[127]. To examine tissue similarity, median gene expression for each tissue was calculated and then used to cluster tissues based on the Euclidean distance with the corresponding function in the R package ComplexHeatmap (v2.8.0)[128].

Then, the 19 whole-body tissues were classified into 10 different broad categories (i.e., muscle, immune, intestine, rumen, brain, artery, adipose, liver, lung and kidney) (Supplementary Data 1). To characterize the genes showing tissue-specific expression, we compared gene expression in the samples of a target tissue with that in tissues from the other categories[129] using the R package limma (v3.48.3)[130]. Known covariates (e.g., age and time point) were calibrated via the *combat* function of the R package sva (v.3.40)[131]. We ranked *t*-statistics computed by limma and considered the top 5% of genes with an absolute value of $\log_2$-transformed fold-change ($|\log_2(FC)|$) > 1 and a false discovery rate (FDR) < 0.01 as tissue-specific genes (TSGs).

For the ATAC-Seq data, we used a Shannon entropy-based method[132] to compute the tissue specificity index with a normalized peak matrix. Specifically, for each peak, we defined its relative accessibility in tissue $i$ as $Ri = \frac{Ei}{\sum_{i=1}^{N} Ei}$, where $E_i$ is the normalized median reads per kilobase million (RPKM) value for the peak in tissue $i$, $\sum_{i=1}^{N} Ei$ is the sum of normalized median RPKM values in all tissues, and $N$ is the total number of tissues. The peak entropy score across tissues can be defined as $H = \sum_{i=1}^{N} -1 * Ri * \log 2(Ri)$, where the value of $H$ ranges between 0 and $\log_2(N)$. An entropy score close to zero indicates highly tissue-specific accessibility of the peak; conversely, a score close to $\log_2(N)$ indicates ubiquitous accessibility of the peak. We selected peaks with an entropy score < 2.5 as the tissue-restricted peaks, while peaks were assigned to particular tissues with maximal RPKM values. Peaks with the top 500 entropy scores were considered conserved peaks.

## Comparative transcriptome analysis
Human RNA-Seq data normalized by the transcripts per million (TPM) method was generated by the human GTEx consortium. From the GTEx v8 release (https://gtexportal.org/home/datasets), we obtained a subset of 6792 RNA-Seq samples from 14 common tissues and 17,279 one-to-one orthologous genes between human and sheep. We then used the function *IntegrateData* with the parameters (anchorset = expression, dims = 1:30) in the R package Seurat (v4.2.0)[133] to combine the expression values of orthologous genes in human and sheep by removing hidden confounding factors. Thereafter, we performed *t*-SNE clustering of the samples using a two-dimensional projection based on corrected expression values of orthologous genes. We calculated the median values of gene expression in each tissue of sheep and human separately, representing the expression of the particular tissue in each species[134]. Additionally, we performed hierarchical clustering using the R package ComplexHeatmap (v2.8.0)[128] to explore the relationships of tissues based on the median values.

## Characterization of gene expression changes along a timeline
In each tissue of the Hu sheep translocated to high altitude, we identified significantly differentially expressed genes (DEGs) between adjacent time points (0 d vs. 7 d, 7 d vs. 14 d, 14 d vs. 21 d, and 21 d vs. ~8 mon) with thresholds of an FDR < 0.05 and $|\log_2(FC)|$ > 0.75 using the R package DESeq2 (v1.32.0)[135]. We also identified DEGs between low-altitude Hu sheep and high-altitude Hu sheep (0 d vs. 7 d, 0 d vs. 14 d, 0 d vs. 21 d and 0 d vs. 8 mon) with aforementioned thresholds and software. For comparisons with great number of DEGs in particular tissues (adjacent time comparisons: "0 d vs. 7 d" in the cerebellum and "7 d vs. 14 d" in the colon; high altitude and low-altitude Hu sheep comparison: "0 d vs. 14 d", "0 d vs. 21 d", "0 d vs. 8 mon" in the cerebellum and "0 d vs. 14 d" in the colon), we used R package powsimR (v0.090)[136] to conduct power analysis and examine whether the

identified great number of DEGs was hypoxic effect on large transcriptional change or technical artifact.

Between the adjacent time points, *k*-means clustering was used to characterize the patterns of gene expression changes in different tissues. We first built a $\log_2$(FC) matrix and determined the *k* value with the *fviz_nbclust* function in the R package factoextra (v1.0.7)[137]. Common genes among different tissues were then clustered into a set of *k* groups based on the Euclidean distance with the *kmeans* function in R.

Additionally, we collected candidate genes related to high-altitude adaptation and mountain sickness in human from previous studies (Supplementary Data 32 and 33). Furthermore, we characterized the time-series patterns of gene expression changes across tissues using the same methods described above.

## Time series analysis

Furthermore, we performed time series analysis of gene expression to identify dynamically changing genes (DCGs) over time using the R package maSigPro (v1.64.8)[138]. First, we calculated the raw count matrix and retained genes with a sum of all counts ≥ 10 reads in all the samples. We then performed normalization with the *vst* function in the R package DESeq2 (v1.32.0)[135]. We ran maSigPro with a degree = 4 and considered genes with a goodness-of-fit ($R^2$) ≥ 0.4 as DCGs. We also applied a soft-clustering approach (*c*-means clustering) in the R package mFuzz (v.2.23.0)[139] to identify the most common profiles for individual tissues. We designated clusters when increasing the number would not add a new cluster but instead split a previous cluster.

## Weighted gene co-expression network analysis (WGCNA)

We employed the R package WGCNA (v1.12.0)[140] to construct a gene co-expression network for an aggregated expression matrix in blood. Similar to the time-series analysis, we kept genes with a sum of all counts ≥ 10 reads in all the samples and a median absolute deviation (MAD) value > 0.01 (top 75% of MAD) and then performed normalization with the *vst* function in the R package DESeq2 (v1.32.0)[135], resulting in 13,707 genes for weighted gene co-expression network analysis. Subsequently, we transformed the normalized matrix to a similarity matrix based on the pairwise Pearson's correlation among the 13,707 genes and then converted the similarity matrix into an adjacency matrix. By using the dynamic hybrid cutting method, we clustered genes with similar expression patterns ($r > 0.85$) into 14 distinct gene modules and used principal component analysis (PCA) to summarize modules of gene expression with the *blockwiseModules* function. We then used module eigengene values of the first principal component to test the correlation between module expression and phenotypic data.

## Genomic annotation of ATAC-Seq peaks

We annotated the ATAC-Seq peaks using the *annotatePeak* function in the R package ChIPseeker (v1.28.3)[141]. Based on the distance from the peak center to the transcription start site (TSS), the peaks were assigned to functional genomic regions such as promoter (TSS ± 2 kb), downstream, distal intergenic, intron, exon, 5′ untranslated region (UTR), and 3′ UTR. We then calculated the percentage of peaks located in the upstream and downstream regions from the TSS of the nearest genes and visualized the distribution using the *plotDistToTSS* function in ChIPseeker.

## Differential accessibility analysis

In each tissue, we defined differentially accessible regions (DARs) among groups (e.g., three groups in dataset 1: the ewes of Hu sheep, the Hu sheep translocated to the QTP after 8 months and Tibetan sheep; three groups in dataset 2: the lambs of Hu sheep, Hu sheep raised on the QTP for -8 months and Tibetan sheep) using the *dba.count, dba.contrast, dba.analyze* and *dba.report* functions of the R

package DiffBind (v3.6.1)[116]. We set the thresholds as follows: *P* value < 0.05 and $|\log_2(FC)| > 0.5$.

## Motif enrichment analysis

We converted the locations of target peaks from the sheep reference genome *Oar_rambouillet_v1.0* (GCA_002742125.1) to the human reference genome *GRCh38* (GCA_000001405.15) using the program LiftOver (v377) with the default settings. Then, enrichment analysis of known binding motifs in peaks was performed using the "findMotifsGenome.pl" script in the software HOMER (v4.8)[142]. *P* values were calculated with a hypergeometric test, and a *P* value < 0.05 was regarded as the threshold for identifying significant motifs.

## Gene-linked candidate *cis*-regulatory elements within TAD

We utilized the correlation approach to identify putative causal relationships between ATAC-Seq peaks and gene expression in the same samples[143,144]. For each TAD in the sheep genome defined above, we computed the Pearson correlation coefficient (PCC) between the estimates of chromatic accessibility [$\log_2$(RPKM + 1)] and gene expression [$\log_2$(TPM + 1)]. To examine the significance level (*P* value) of these correlations, a null distribution was estimated empirically by calculating the PCC of the TAD-constrained peaks with all the genes on the chromosome. Significant associations between TAD-constrained peaks and the expression of relevant genes were identified according to a *P* value < 0.05 and PCC ≥ 0.25.

## Functional enrichment analysis

Human homologs of the sheep Ensembl genes were fetched using the R package biomaRt (v2.52.0)[145]. Gene Ontology (GO) enrichment analysis was implemented based on the human database org.Hs.eg.db (v3.16)[146] to identify significant biological functions using the R package ClusterProfiler (v4.0.5)[147], with the parameters of OrgDB = org.Hs.eg.db, fun = enrichGO, ont = BP, pvalueCutoff = 0.05, and pAdjustMethod = BH.

## Phenome-wide association analysis (PheWAS)

The GWAS ATLAS database (https://atlas.ctglab.nl/) provides abundant resources for associating a given gene or SNP with a wide variety of human phenotypes; this strategy has been widely used and proven to be a useful approach, complementary to regular GWAS[148,149]. To explore whether human orthologues of key candidate genes detected in the context of hypoxia acclimatization in sheep are associated with similar adaptive traits in human, we performed PheWAS analysis for human orthologous genes across 3302 human phenotypes. We included only human GWASs with a sample size > 10,000 in the analysis and considered genes with an FDR < 0.05 to be significantly associated with the corresponding phenotypic trait.

## Analysis of phenotypic measures

For the SpO$_2$ and 19 additional blood biochemical indices, phenotypic measurement data were represented as the mean ± standard error (SE). Statistical analysis was implemented in RStudio v4.2.0. First, we assessed the normality of the dataset with the *shapiro.test* function. Then, statistically significant differences among groups of animals at different time points were determined by one-way ANOVA (analysis of variance) with the function *aov* when the datasets conformed to the normal distribution, and Tukey's honestly significant difference (HSD) test was performed to correct for multiple comparisons using the *TukeyHSD* function. For the datasets not under a normal distribution, the Kruskal-Wallis test was used to estimate the significant differences among groups with the function *kruskal.test*, while the Wilcoxon rank test was used to determine differences between pairwise groups with the *pairwise.wilcox.test* function. Variations of each phenotypic indicator were considered statistically significant with a Benjamini-Hochberg adjusted *P* value < 0.05.

## Analysis of scRNA-Seq data

After sequencing, the high-quality reads were aligned to the sheep reference genome *ARS-UI_Ramb_v2.0* (GCA_016772045.1). We created the premRNA reference following the protocol of 10× Genomics and earlier studies[150]. To obtain the filtered count matrix for subsequent analysis, we calculated gene counts using Cell Ranger (v7.0) (https://github.com/10XGenomics/cellranger) with the default parameters.

The single-cell expression matrices obtained as described above were processed following the protocols of the R package Seurat (v.4.2.0)[133]. We retained cells with more than 200 genes and a low percentage (< 30%) of UMIs mapped to mitochondrial genes. We also used the R package DoubletFinder (v.2.0.3)[151] to mitigate potential doublets with the *doubletFinder_v3* function. We applied the canonical correlation analysis (CCA) method for dataset integration to correct batch effects as follows. First, normalization for each sample was implemented with the *SCTransfrom* function. We then determined features and anchors for data integration using the *PrepSCTIntegration* and *FindIntegrationAnchors* functions with the default parameters. Furthermore, we generated an integrated expression matrix for each tissue via the *IntegrateData* function.

We scaled the integrated data and applied the *RunPCA* function in the dimensional reduction. Cells were clustered with the *FindClusters* function and visualized using uniform manifold approximation and projection (UMAP). Gene markers for each cell type or subtype were found using *FindAllMarkers* with the Wilcoxon rank sum test ($P < 0.05$, |$\log_2 FC$| > 0.25), including known canonical marker genes and novel ones (Supplementary Data 37).

We predicted core regulatory transcription factor (TF)-gene pairs from the scRNA-Seq data using the R package GENIE3 (v1.6.0)[152] with the RcisTarget database v1.4.0 (https://resources.aertslab.org/cistarget/) as a reference. Specifically, we used GENIE3 and the workflow in R package SCENIC (v1.1.2.2)[153] with the default parameters to infer TF-target gene regulatory networks based on the gene expression matrices of each tissue and the DEGs of each cell type. Then, RcisTarget was used to identify enriched TF-binding motifs and predict candidate target genes (regulons) based on the hg38 RcisTarget database, which contains cross-species genome-wide rankings for each motif. Only the TF target genes with high-confidence annotations and corresponding transcription regulatory networks were visualized with Cytoscape (v3.7.1)[154].

Cell-cell communication was inferred from the scRNA-Seq data using CellPhoneDB software (v2.0)[155]. Only receptors and ligands expressed in at least 10% of cells of a given cell type were retained for further analysis, whereas the interaction was considered nonexistent if either the ligand or the receptor was unqualified. The average expression of each ligand–receptor pair was compared among different cell types to identify their cell-specific expression. Only the ligand–receptor pairs with significant means ($P < 0.05$) were used for subsequent inference of cell-cell communications in each tissue across time points. The visualization of the changes in the number of ligand–receptor pairs along the examined timescale for each tissue was implemented using Cytoscape (v3.7.1)[154].

## Statistics and reproducibility

No statistical method was used to predetermine the sample size, and no data were excluded from the analyses. The experiments were not randomized, and the investigators were not blinded to allocation during experiments and outcome assessment.

For all the boxplots, the horizontal lines inside the boxes represent the median or mean values. Box bounds indicate the lower quartile (Q1, the 25th percentile) and the upper quartile (Q3, the 75th percentile). Whiskers represent minima (Q1 − 1.5 × IQR) and maxima (Q3 + 1.5 × IQR), where IQR is the interquartile range (Q3-Q1). The values of each individual were plotted with data points.

## Reporting summary

Further information on research design is available in the Nature Portfolio Reporting Summary linked to this article.

## Data availability

The high-throughput sequencing data generated in this study have been deposited in the Sequence Read Archive (SRA) database under accession code PRJNA1053506 for WGS, PRJNA1000743 and PRJNA1001016 for RNA-Seq and PRJNA1001505 for ATAC-Seq and scRNA-Seq data. The public data used in this study are available in the SRA database under accession code SRR19426890 for Hi-C and PRJNA624020, PRJNA645671 and PRJNA160933 for WGS data. For the WGS data from Lv et al. [30], access can be obtained by contacting the author of that paper for research purposes. The processed RNA-Seq data are available at Gene Expression Omnibus (GEO) database under accession code GSE261409. Source data are provided with this paper.

## Code availability

All the computational scripts and codes used in this study are available on GitHub repository: https://github.com/zeyan-0717/sheep-transcriptome-atlas and Zenodo: 10.5281/zenodo.10799788.

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

## Acknowledgements

This study was financially supported by grants from the National Key Research and Development Program of China (Nos. 2021YFD1200900 to M.H.L., 2022YFE0113300 and 2021YFF1000703 to J.Y., and 2021YFD1300904 to F.H.L.), and the National Natural Science Foundation of China (Nos.32320103006 to M.H.L., 32272845 to J.Y., and 32061133010, 31972527 and U21A20246 to F.H.L.). We thank all synonymous reviewers for their valuable comments on the peer review of this work. We also thank the High-performance Computing Platform of China Agricultural University.

## Author contributions

M.H.L. conceived and supervised the study. M.H.L. and Z.Y. designed the project proposal. Z.Y. performed bioinformatic analysis of multi-omics data. F.H.L. conducted whole-genome sequencing data analysis, and W.T.W. performed single-cell RNA-Seq data analysis. Z.Y., M.L.Z., Y.J.L., D.X.M., W.T.W., R.M., X.W., M.M.W. and J.H.H. contributed to the sample collection and resource generation. Z.Y., J.Y., and M.H.L. wrote and revised the manuscript. All authors reviewed, edited, and approved the final manuscript.

## Competing interests

The authors declare no competing interests.
