## [Peer Review File · Nature Communications]

REVIEWER COMMENTS

Reviewer #1 (Remarks to the Author):

This is an interesting study of transcriptional changes and biochemical altitude adaptation that compares ewes from low altitude plains, the same breed moved to high altitude, and a different breed at high altitude that are already well adapted to high altitude. Moreover, they examined offspring to look for effects of altitude during pregnancy. This is a unique and valuable study.

The methods are appropriate and carried out to a high standard.

The figures are clear, and helpful.

The interpretation of the transcriptional changes is arguably speculative in some places, but in my view it is not over done, and the point of the study is to document the changes.

“To test the heritability of acclimatization to hypoxia,”. I recommend being more cautious here to avoid leaping to conclusions. The transplanted lambs were exposed to hypoxia in utero. Thus, it is not possible to distinguish effects due to direct antenatal exposure to low partial pressures of oxygen on the fetus and a true heritable effect. You may wish to rephrase the heading and discussion as ‘Acclimatisation to high altitude in offspring’ or similar.

The Ms is generally very well written and clear.

There are just a few unclear redactory issues:

e.g.

“ewes of Hu sheep the plain” ? did you mean “Hu sheep ewes from low altitude plains” or similar?

Similarly; would “ewes of Tibetan sheep on the plateau” better as “Tibetan sheep at high altitude” or similar?

In general, it might be worth considering using high altitude rather than plateau and low altitude rather than plain, through out the MS??

Figures; many journals recommend presenting data in bar graphs as vertical scatter plots, so that the individual data are shown.

Reviewer #2 (Remarks to the Author):

In this report Yan et al provide an extensive investigation on how sheep respond to hypoxia (altitude changes) at the physiological and genomic level. The amount of data provided, when made public will be an amazing resource for the community including basic and medical and veterinarian research related.

In terms of novelty the data is completely novel, however given the sheer amount of data provide, there is not a message being stated. It is very difficult to extract what is important from what is just more information, so the main message is missing. Some specific points below:

1. As this paper looks at gene transcription changes in hypoxia, it is important to mention the HIF-PHD-VHL pathway in the introduction. Furthermore, authors show changes in HIF1A gene expression and mention that “HIF1A, associated with high-altitude hypoxia adaptation^{4,81}”. Whilst the EPAS1(HIF-2 α) gene and EGLN (PHD) genes are associated with high-altitude hypoxia adaptation I am not sure if the HIF1A gene is. Typically HIF-1 α protein is stabilized in hypoxia via dioxygenase inhibition, it is unclear if the authors are stating that changes in HIF1A gene expression are associated with high-altitude hypoxia adaptation, as the 2 references they give don't support this. This point needs clarification. It is also important to mention EPAS1 and EGLN genes in the introduction, regarding high-altitude hypoxia adaptation.
2. Relating to point 1, authors should show what happens to the expression and accessibility of other the other HIF genes (ARNT and EPAS1), and dioxygenases, particularly the PHD (EGLN) genes, in the adaption to high altitude in their model. Authors should also see if expression of HIF and PHD genes across different tissues in plain sheep at low altitude correlates/anti-correlates/predicts the speed/extent to which different tissues respond to hypoxia when these sheep and transplanted to high altitude. Also does the expression of these genes differ between Tibetan and plain sheep?
3. Figure 5 e,f. Authors show that the majority OF genes with both differential expression and differential chromatin accessibility during hypoxia acclimatization are downregulated, rather than upregulated, and discuss this very briefly in the discussion. I feel this is a striking and important finding, highlighting the importance of chromatin dynamics in response to hypoxia, particularly with regard to gene repression. The molecular mechanisms dictating gene repression in hypoxia are less well understood than gene activation. The findings are figure 5 e,f should be discussed more in the discussion section, with supporting references.
4. A further point regarding figure 5, there is literature showing HIF target gene/hypoxia responsive gene selection across different cell types is, at least part, determined by the chromatin landscape. Does differences in chromatin accessibility across tissues in plain sheep at low altitude explain/correlate with differences in differentially expressed genes across tissues in response to the high altitude transplantation?
5. does the chromatin change similar across different tissues or is it very tissue specific. There is some data on the extended version but this is not explained or placed into context.

Reviewer #3 (Remarks to the Author):

This study tackles the biologically and agronomically relevant question of genetic and phenotypically plastic adaptations to high altitude in sheep. The authors collect an impressive amount of data (e.g. 1277 RNA-seq, 66 ATAC-seq libraries and 6 scRNA-seq libraries easily worth >100k\$). Even more importantly, the experiment is well designed, allowing to contrast short term and long term adaptations of lowland sheep to high altitudes and allowing to put the molecular data in context of 20 physiological measurements. In addition, they add genomic data and Hi-C data from previous papers. Together, this makes indeed a “comprehensive time-series transcriptome atlas” in response to hypoxia.

The resulting possibilities in analyses are enormous and each section of the results could be an own paper (and maybe should be as some of the analyses are rather ad hoc and superficial and in its entirety not “reviewable”). This said, the authors do an ok job for a first analysis and the paper is still in principle very useful as a resource for future analyses.

However, I have a few major issues that would need to be addressed to improve the impact of the paper.

Major issues:

1) The most important contribution of the paper is to provide a comprehensive atlas, but for such an atlas to be useful, the data and metadata need to be much more accessible. Currently, it is just available in the Short Read archive and I could not access to data there to check the quality of the metadata. I think that metadata is generally much more comprehensive in GEO, so I would suggest to deposit data there. In any case, reviewers or editors should have access to check whether the metadata is sufficiently clear. In addition, the authors should provide the basic scripts, data tables, genome annotations and resulting result files at a repository like GitHub.

2) The comparison of DEG across tissues and timepoints is very interesting, as it provides in principle an unbiased assessment of which tissue is especially affected when after transfer to high altitudes (In general, I find this section of the paper better and more important than the following ones). However, in order to be able to compare the number of DEGs across tissues and timepoints, they need to have comparable power to detect DEGs. This depends especially on the number of replicates (which one would need to dig out from the supplementary tables) but also on the variation among samples of the same timepoints and tissues. So the authors need to show that e.g. the large effect on the cerebellum in the first seven days or the big effect on colon from 7 to 14 days is not a technical artifact of power. An ideal solution would be to do power analyses using tools like powsimR or RNASeqPower. I think this part could also be extended a bit more by describing better how the magnitude of change develops across tissues and time not only in pairwise day comparisons, but also in relation to d0

3) The combination of FST and DEG and DCG genes is interesting, but very ad hoc and not well justified (beyond the urge to identify few genes ...). In addition, it is not clear whether the observed overlap is significant. Permutations of DEG or FST genes could help here

I have compiled a non-comprehensive number of small issues that I think could be improved

1) I am confused by the term “plain-to-plateau transplantation” shouldn’t it be rather “plain-to-plateau translocation”? I am not a native speaker, but I think that transplantation should be replaced by translocation throughout the manuscript.

2) I think the headings of the subsections should be more informative and consistent

3) I think it is more helpful to report average values with confidence intervals in the text when reporting e.g. SpO2 differences rather than two digit precise very low p-values

4) L. 168: “significantly correlated (FDR < 0.05) with the bio-indicators” should be “significantly correlated (FDR < 0.05) with one or more bio-indicator”

5) Line 158ff “..only observed a typical pattern of gene expression in the abomasum, which showed an obvious separation of samples before and after 14 d” what is a typical pattern? This is unprecise/unclear

6) At the beginning of the section “Regulation of gene expression by TAD-constrained cis-regulatory elements” it should be made clearer that the authors used previous Hi-C data to assign TADs and from which tissue the TADs were constructed.

Reviewer #4 (Remarks to the Author):

This work examined multi-omics changes of sheep in response to high-altitude stress. The experiments are complete and all analyses are good. Data and cost are surprising. I had thought to use cattle to do the similar analyses and did not do because of the lack of the enough founding. Overall, I am satisfied with experiments designed and the data collected. There are several questions that need to be addressed.

1. Which breed or group is sister or closely related to Tibet Sheep? Is it Hu Sheep? If not, how is the phylogenetic relationship will affect the identified genes during the low-term evolution? I suggest to use alternative sheep breed to examine such a divergence.

2. According to my knowledge, some Hu sheep also occur in the high-altitude region in Sichuan. How do you determine that the Hu sheep materials you used have not been experienced the high-altitude pre-adaptation? Is it possible to give a phylogenetic relationship of the used Hu sheep and those in the other regions?

3. Have Tibet sheep experienced introgression and obtained the related genes from other sheep breeds that had acclimated to the Tibet high-altitude regions earlier? How did you exclude this possibility? If not, how will this will affect your conclusion?

4. Tibet sheep may also obtain their high-altitude variants from the ancestral polymorphisms before their occurrences in the Tibet. They may have obtained such variants through interspecific introgressions. How do you exclude this possibility?

5. For high-altitude adaptation, reproductive survivals are more important than others, especially for humans. Female Tibetans with Hif1A variants are more likely producing the alive babies when without any modern care. Is it possible to add these reproductive phenotypes when compared them?

Reviewers' comments

Reviewer #1 (Remarks to the Author)

This is an interesting study of transcriptional changes and biochemical altitude adaptation that compares ewes from low altitude plains, the same breed moved to high altitude, and a different breed at high altitude that are already well adapted to high altitude. Moreover, they examined offspring to look for effects of altitude during pregnancy. This is a unique and valuable study. The methods are appropriate and carried out to a high standard. The figures are clear, and helpful.

Thank you for your positive comments.

The interpretation of the transcriptional changes is arguably speculative in some places, but in my view it is not over done, and the point of the study is to document the changes. "To test the heritability of acclimatization to hypoxia,". I recommend being more cautious here to avoid leaping to conclusions. The transplanted lambs were exposed to hypoxia in utero. Thus, it is not possible to distinguish effects due to direct antenatal exposure to low partial pressures of oxygen on the fetus and a true heritable effect. You may wish to rephrase the heading and discussion as 'Acclimatisation to high altitude in offspring' or similar.

[1.1] Good comment! Following your suggestion, we have changed the relevant content of "the heritability of acclimatization to hypoxia" to "acclimatization to high altitude in offspring" or phrases with similar meaning. We also added a brief discussion to admit that this study can't distinguish whether the effects were due to direct antenatal exposure to low oxygen in utero or a true heritable effect. Please see **lines 664-667**.

Lines 664-667:

"Apart from the aforementioned tissue-level regulatory mechanism of high altitude Hu lamb, it should be noted that at current time we can't distinguish whether the successful

acclimatization of Hu lamb was due to direct antenatal exposure to low oxygen in utero or a true heritable effect.”

The Ms is generally very well written and clear. There are just a few unclear redactory issues:

e.g. “ewes of Hu sheep the plain” ? did you mean “Hu sheep ewes from low altitude plains” or similar?

[1.2] We have changed as “Hu sheep ewes from low altitude” according to your suggestion.

Similarly; would “ewes of Tibetan sheep on the plateau” better as “Tibetan sheep at high altitude” or similar?

[1.3] We have changed as “Tibetan sheep ewes at high altitude” according to your suggestion.

In general, it might be worth considering using high altitude rather than plateau and low altitude rather than plain, through out the MS??

[1.4] We think it is better to state the specific plateau and plain that the animals originally inhabited at the beginning of experimental design. We used high altitude and low altitude to replace plateau and plain, respectively, in the other parts of the manuscript as suggested.

Figures; many journals recommend presenting data in bar graphs as vertical scatter plots, so that the individual data are shown.

[1.5] We have updated all the bar graphs with vertical scatter plots to show the individual data directly. Please see **Fig. 2a, Fig. 5h and Fig. 6a, e.**

Reviewers' comments

Reviewer #2 (Remarks to the Author):

In this report Yan et al provide an extensive investigation on how sheep respond to hypoxia (altitude changes) at the physiological and genomic level. The amount of data provided, when made public will be an amazing resource for the community including basic and medical and veterinarian research related.

In terms of novelty the data is completely novel, however given the sheer amount of data provide, there is not a message being stated. It is very difficult to extract what is important from what is just more information, so the main message is missing. Some specific points below:

1. As this paper looks at gene transcription changes in hypoxia, it is important to mention the HIF-PHD-VHL pathway in the introduction. Furthermore, authors show changes in HIF1A gene expression and mention that "HIF1A, associated with high-altitude hypoxia adaptation". Whilst the EPAS1(HIF-2 α) gene and EGLN (PHD) genes are associated with high-altitude hypoxia adaptation I am not sure if the HIF1A gene is. Typically HIF-1 α protein is stabilized in hypoxia via dioxygenase inhibition, it is unclear if the authors are stating that changes in HIF1A gene expression are associated with high-altitude hypoxia adaptation, as the 2 references they give don't support this. This point needs clarification. It is also important to mention EPAS1 and EGLN genes in the introduction, regarding high-altitude hypoxia adaptation.

[2.1] Good comments! We mentioned *EPAS1* and *EGLN* genes and the HIF-PHD-VHL pathway in the introduction as suggested (**lines 40-42**: "*such as EPAS1 and EGLN genes (e.g., EGLN1, EGLN2, EGLN3) in the hypoxia-inducible factor (HIF)-prolyl hydroxylase domain (PHD)-Von Hippel-Lindau (VHL) pathway*"). Regarding *HIF1A*, we intended to state that changes in its gene expression are associated with high-altitude hypoxia adaptation, and the references 4 and 81 were not properly cited. In the revised version, we replaced the previous references with new ones (**references 88 and 89**) which can

support our statement of *HIF1A*. Please see **lines 683-684** (“changes in gene expression are associated with high-altitude hypoxia adaptation^{88,89}”) in the revised manuscript.

References

88. Martinez, C.-A. et al. Intermittent hypoxia enhances the expression of hypoxia inducible factor HIF1A through histone demethylation. *J. Biol. Chem.* 2022;298(11):102536.
89. Bai, J., Li, L., Li, Y. & Zhang, L. Genetic and immune changes in Tibetan high-altitude populations contribute to biological adaptation to hypoxia. *Environ. Health Prev. Med.* 27, 39 – 39 (2022).

2. Relating to point 1, authors should show what happens to the expression and accessibility of the other HIF genes (*ARNT* and *EPAS1*), and dioxygenases, particularly the PHD (*EGLN*) genes, in the adaption to high altitude in their model. Authors should also see if expression of HIF and PHD genes across different tissues in plain sheep at low altitude correlates/anti-correlates/predicts the speed/extent to which different tissues respond to hypoxia when these sheep and transplanted to high altitude. Also does the expression of these genes differ between Tibetan and plain sheep?

[2.2] Following the comments, we examined the expression and accessibility of the three HIF genes (i.e., *HIF1A*, *ARNT* and *EPAS1*) and a PHD gene (i.e., *EGLN2*, belong to the *EGLN* gene family). For each tissue, we showed the expression of these genes in high altitude Hu sheep at four acclimatization time points, and evaluated the statistical significance for the differences in gene expression between high altitude and low altitude Hu sheep (**Revised Extended Data Fig. 21**). We found that changes in expression of these genes in particular tissues of our sheep model showed similar trends across four acclimatization time points. For example, when compared to low altitude Hu sheep, consistently and significantly ($P < 0.05$, Wilcoxon rank sum test) up-regulated expression of *HIF1A*, *EPAS1*, *ARNT* and down-regulated expression of *EGLN2* were observed in the cerebellum in high altitude Hu sheep (**Revised Extended Data Fig. 21**). Also, the expression of *EPAS1*, *ARNT* and *EGLN2* in blood was

significantly ($P < 0.05$, Wilcoxon rank sum test) decreased (**Revised Extended Data Fig. 21c, d**). Please see **lines: 491-497**.

Lines 491-497:

“As the HIF and PHD genes are key regulators of high altitude adaptation, we further examined the expression patterns of three HIF genes (i.e., EPAS1, ARNT and HIF1A) and a PHD gene (i.e., EGLN2) in our sheep model. We found consistently and significantly (Wilcoxon rank sum test, $P < 0.05$) up-regulated expression of HIF genes EPAS1, ARNT and HIF1A and down-regulated expression of PHD gene EGLN2 in cerebellum in high altitude Hu sheep (i.e., 7 d, 14 d, 21 d and 8 mon) as compared to low altitude Hu sheep (Extended Data Fig. 21).”

We then examined the accessibility of the above four genes in low altitude Hu sheep and high altitude Hu sheep after 8 month translocation for each tissue with Integrative Genomics Viewer (IGV) (**Fig. R1**). We found an overall higher accessibility of EPAS1 and EGLN2 than those of HIF1A and ARNT. For each gene, we observed tissue specificity of accessibility and differences in accessibility between sheep groups within the same tissue. For example, EPAS1 was more accessible in the lung, heart and liver than in the hypothalamus, spleen and adipose. Within the liver, the peak density of EPAS1 in low altitude Hu sheep was higher than that in high altitude Hu sheep.

Fig. R1 Chromatin accessibility for HIF genes (*HIF1A*, *EPAS1*, *ARNT*) and PHD gene (*EGLN2*) between low altitude Hu sheep and high altitude Hu sheep across tissues.

We calculated Pearson's correlation coefficient for the expression of the above four genes across tissues between low altitude Hu sheep (i.e., 0 d) and high altitude Hu sheep (i.e., 7 d, 14 d, 21 d and 8 mon) (**Revised Extended Data Fig. 22**). Overall, *EPAS1*, *EGLN2* and *ARNT* exhibited significant ($P < 0.05$) correlation coefficients, with *EPAS1* showing the highest values ($r > 0.97$) (**Revised Extended Data Fig. 22b-d**). For individual tissue, we found that the expression of three *HIF* genes *EPAS1*, *ARNT* and *HIF1A* in the cerebellum showed the maximum extent of expression changes across the four acclimatization time points when low altitude Hu sheep were translocated to high altitude (**Revised Extended Data Fig. 22a-c**). Please see **lines 497-504**.

Lines 497-504:

“The expression of EPAS1, EGLN2 and ARNT across tissues in low altitude Hu sheep was significantly ($P < 0.05$) correlated with that in high altitude Hu sheep, and EPAS1 showed the highest correlation coefficient values (Pearson’s $r > 0.97$) (Extended Data Fig. 22). For individual tissues, the expression of HIF genes EPAS1, ARNT and HIF1A in cerebellum showed the maximum extent of expression changes across the four acclimatization time points when low altitude Hu sheep were translocated to high altitude (Extended Data Fig. 21a-c).”

We also compared the expression of the above four genes between low altitude Hu sheep and Tibetan sheep across tissues (**Revised Extended Data Fig. 23**). We observed that the expression levels of HIF genes HIF1A, EPAS1 and ARNT in the cerebellum were significantly higher ($P < 0.05$, Wilcoxon rank sum test) in Tibetan sheep than in low altitude Hu sheep. This was similar to the higher expression of the three HIF genes captured in the cerebellum in high altitude Hu sheep as described above. These results indicated that high expression levels of HIF genes in the cerebellum could have played an important role in the acclimatization and adaptation to hypoxia. Please see **lines 504-508**.

Lines 504-508:

“We also observed that the expression levels of HIF genes EPAS1, ARNT and HIF1A in the cerebellum were significantly (Wilcoxon rank sum test, $P < 0.05$) higher in Tibetan sheep than those in low altitude Hu sheep (Extended Data Fig. 23). These results indicate that the expression patterns of HIF genes in the cerebellum may play an important role in the acclimatization and adaptation to hypoxia.”

3. Figure 5 e,f. Authors show that the majority OF genes with both differential expression and differential chromatin accessibility during hypoxia acclimatization are downregulated, rather than upregulated, and discuss this very briefly in the discussion. I feel this is a striking and important finding, highlighting the importance of chromatin dynamics in response to hypoxia, particularly with regard to gene repression. The molecular mechanisms dictating gene repression in hypoxia are less

well understood than gene activation. The findings are figure 5 e,f should be discussed more in the discussion section, with supporting references.

[2.3] Good comments! We agree with you that oxygen sensing machinery (e.g., hypoxia response) can be subjected to both positive (e.g., gene activation or up-regulation) and negative regulatory (e.g., gene repression or down-regulation) feedbacks. Following your suggestions, we provided specific examples of Tibetans (**reference 79**) and deer mice (**reference 78**) to support our results, which demonstrated the importance of down-regulation of high-altitude adaptation associated genes in response to hypoxia. We also highlighted the important findings relevant to whole-genome chromatin dynamics and associated gene repression in response to hypoxia, and added the implications of these findings. We have added the above contents and supporting references in the relevant part of discussion, please see **lines 618-623 and 636-641**.

References

78. Storz, J. F. & Cheviron, Z. A. Functional Genomic Insights into Regulatory Mechanisms of High-Altitude Adaptation. in *Hypoxia: Translation in Progress* (eds. Roach, R. C., Hackett, P. H. & Wagner, P. D.) 113–128 (Springer US, 2016).
79. Xin, J., Zhang, H., He, Y. et al. Chromatin accessibility landscape and regulatory network of high-altitude hypoxia adaptation. *Nat. Commun.* 11, 4928 (2020).

Lines 618-623:

“Oxygen sensing machinery can be subject to both positive (e.g., up-regulation) and negative regulatory (e.g., down-regulation) feedbacks⁷⁷. Our results were consistent with previous findings showing that negative feedback loops, e.g., down-regulation of angiogenesis associated genes (e.g., PPARG) in deer mice and down-regulation of high-altitude adaptation genes (e.g., EPAS1) in Tibetans, play an important role under hypoxic stress^{78,79}.

Lines 636-641:

“With regard to the chromatin and expression changes within high altitude Hu sheep, we provided an exceptional example for whole-genome chromatin dynamics in response to hypoxia with chromatin accessibility mostly decreased by down-regulation, which subsequently repressed the expression of most genes. The above findings could contribute to our understanding of the molecular mechanisms dictating gene repression in hypoxia at the transcriptomic and epigenomic level.”

4. A further point regarding figure 5, there is literature showing HIF target gene/hypoxia responsive gene selection across different cell types is, at least part, determined by the chromatin landscape. Does differences in chromatin accessibility across tissues in plain sheep at low altitude explain/correlate with differences in differentially expressed genes across tissues in response to the high altitude transplantation?

[2.4] Following your comments, we found that several papers (**references 80-82**) used different cell lines (e.g., HeLa cells, HL-1 cells and HUVEC cells) to investigate hypoxia-induced changes to chromatin accessible regions and associated gene expressions. These earlier studies revealed the impact of chromatin landscape on gene selection and gene regulation across different cell types. In this study, we evaluated the chromatin accessibility across tissues in low altitude Hu sheep and its correlation with the expression of differentially expressed genes across tissues in translocated high altitude Hu sheep by linear regression. In detail, we first conducted differential expression analysis and identified differentially expressed genes (DEGs) between low altitude Hu sheep (i.e., 0 d) and translocated high altitude Hu sheep at each time point (i.e., 7 d, 14 d, 21 d and 8 mon) for the tissues having both RNA-seq and ATAC-Seq data. We subsequently extracted open accessibility regions (OARs) in low altitude Hu sheep which are located in the DEGs from each of the above comparisons.

References

80. Wang, J., Wang, Y., Duan, Z. & Hu, W. Hypoxia-induced alterations of transcriptome and chromatin accessibility in HL-1 cells. *IUBMB Life* 72, 1737–1746 (2020).
81. Batie, M., Frost, J., Shakir, D. & Rocha, S. Regulation of chromatin accessibility by hypoxia and HIF. *Biochemical Journal* 479, 767–786 (2022).
82. Moore, L. G. Fetal Growth Restriction and Maternal Oxygen Transport during High Altitude Pregnancy. *High Altitude Medicine & Biology* 4: 141 – 156 (2003).

We then calculated the average value of peak signal for extracted OARs and the expression levels for DEGs for each comparison, and used linear regression to evaluate the correlation between them across tissue (**Revised Extended Data Fig. 28**) by R (4.3.2.0) “*lm*” function. We found that chromatin accessibility in low altitude Hu sheep had moderate correlation ($r = 0.33 - 0.49$, $P > 0.05$) with the expression of DEGs in high altitude Hu sheep at the four acclimatization time points (7 d, 14 d, 21 d and 8 mon) across tissues (**Revised Extended Data Fig. 28**). The results suggest that differences in OARs across tissues in low altitude Hu sheep might partially explain differences in expression of DEGs across tissues in translocated high altitude Hu sheep. Please see **lines 623-636**.

Lines 623-636:

“Additionally, previous studies investigated hypoxia-induced changes to chromatin accessible regions and associated gene expressions, and revealed the impact of chromatin landscape on gene regulation across different cell types (e.g., HeLa cells, HL-1 cells and HUVEC cells)⁷⁹⁻⁸¹. Here, we evaluated the chromatin accessibility in low altitude Hu sheep and its correlation with the expression of differentially expressed genes in translocated high altitude Hu sheep across tissues. Results from the liner regression between the average values of peak signals and gene expression levels showed that chromatin accessibility in low altitude Hu sheep had low to moderate correlations ($r = 0.33 - 0.49$, $P > 0.05$) with the expression of DEGs in high altitude Hu sheep at the four acclimatization time points (7 d, 14 d, 21 d and 8 mon) across tissues (Extended Data Fig. 28). This implied that differences in OARs across tissues in low altitude Hu

sheep did not have significant impact on the expression of DEGs across tissues in translocated high altitude Hu sheep.”

Additionally, we also calculated the correlation between peak signal of OARs in low altitude Hu sheep and expression of DEG in translocated high altitude Hu sheep for each tissue using the median peak and expression values from the individuals in the different Hu sheep groups. The results showed weak correlation ($r < 0.2$, $P > 0.05$) between chromatin accessibility in low altitude Hu sheep and the expression of DEGs in high altitude Hu sheep at most cases for each tissue. Please see **Fig. R2**

Fig. R2 Linear regression between the peak signal value of low altitude Hu sheep and the expression of DEGs from low altitude Hu sheep and high altitude Hu sheep in different comparisons for each tissue. Median values were used for regression. P values were evaluated by Pearson correlation. No peaks located in DEGs from low altitude Hu sheep and high altitude Hu sheep (7 d) in the hypothalamus.

5. does the chromatin change similar across different tissues or is it very tissue specific. There is some data on the extended version but this is not explained or placed into context.

[2.5] Thank you for your comments. We checked the “Extended Data” file and found that **Revised Extended Data Fig. 18** exhibited some relevant results for tissue-specific and conserved peaks. In fact, the chromatin change (e.g., peak) similar or specific across different tissues is associated with its genomic location. Peaks located in the distal intergenic and intron regions are more tissue-specific, whereas those in the promoter, exon, 3' UTR, 5' UTR and downstream regions are more conserved (**Revised Extended Data Fig. 13a, b**). We have added the results in the “*Characterization of chromatin accessibility across tissues under hypoxia*” section. Please see **lines 357-361** in the revised version.

Lines 357-361:

“We characterized tissue-specific and conserved peaks across tissues. The results showed that peaks located in the distal intergenic and intron regions are more tissue-specific, whereas those in the promoter, exon, 3' UTR, 5' UTR and downstream regions are more conserved (Extended Data Fig. 18a, b)”

Reviewers' comments

Reviewer #3 (Remarks to the Author):

This study tackles the biologically and agronomically relevant question of genetic and phenotypically plastic adaptations to high altitude in sheep. The authors collect an impressive amount of data (e.g. 1277 RNA-seq, 66 ATAC-seq libraries and 6 scRNA-seq libraries easily worth >100k\$). Even more importantly, the experiment is well designed, allowing to contrast short term and long term adaptations of lowland sheep to high altitudes and allowing to put the molecular data in context of 20 physiological measurements. In addition, they add genomic data and Hi-C data from previous papers. Together, this makes indeed a “comprehensive time-series transcriptome atlas” in response to hypoxia.

The resulting possibilities in analyses are enormous and each section of the results could be an own paper (and maybe should be as some of the analyses are rather ad hoc and superficial and in its entirety not “reviewable”). This said, the authors do an ok job for a first analysis and the paper is still in principle very useful as a resource for future analyses. However, I have a few major issues that would need to be addressed to improve the impact of the paper.

Major issues:

1) The most important contribution of the paper is to provide a comprehensive atlas, but for such an atlas to be useful, the data and metadata need to be much more accessible. Currently, it is just available in the Short Read archive and I could not access to data there to check the quality of the metadata. I think that metadata is generally much more comprehensive in GEO, so I would suggest to deposit data there. In any case, reviewers or editors should have access to check whether the metadata is sufficiently clear. In addition, the authors should provide the basic scripts, data tables, genome annotations and resulting result files at a repository like GitHub.

[3.1] Thanks for your comments. We agree with you that metadata is crucial to data reusability and reproducibility and facilitates secondary analysis for novel discoveries. We have released our data and metadata and deposited them in NCBI, which are fully accessible to the public. Please check the details of the data with the hyperlinks below:

Metadata: <https://github.com/zeyan-0717/sheep-transcriptome-atlas/blob/main/Data%20table.xlsx>;

WGS: <https://www.ncbi.nlm.nih.gov/sra/?term=PRJNA1053506>;

Bulk RNA-Seq: <https://www.ncbi.nlm.nih.gov/sra/?term=PRJNA1000743>,
<https://www.ncbi.nlm.nih.gov/sra/?term=PRJNA1001016>;

ATAC-Seq and scRNA-Seq: <https://www.ncbi.nlm.nih.gov/sra/?term=PRJNA1001505>.

In fact, sequencing data uploaded to GEO were deposited in SRA eventually, so we think it might be not necessary to upload data repeatedly to GEO. In addition, we have uploaded our basic scripts, data tables, genome annotations and resulting result files at GitHub repository: <https://github.com/zeyan-0717/sheep-transcriptome-atlas>. Please check the details.

2) The comparison of DEG across tissues and timepoints is very interesting, as it provides in principle an unbiased assessment of which tissue is especially affected when after transfer to high altitudes (In general, I find this section of the paper better and more important than the following ones). However, in order to be able to compare the number of DEGs across tissues and timepoints, they need to have comparable power to detect DEGs. This depends especially on the number of replicates (which one would need to dig out from the supplementary tables) but also on the variation among samples of the same timepoints and tissues. So the authors need to show that e.g. the large effect on the cerebellum in the first seven days or the big effect on colon from 7 to 14 days is not a technical artifact of power. An ideal solution would be to do power analyses using tools like powsimR or RNASeqPower. I think this part could also be extended a bit more by describing better how the magnitude of change develops across tissues and time not only in pairwise day comparisons, but also in relation to d0.

[3.2] Good comments again! We implemented power analysis for each of the differential expression analyses with powsimR software as suggested (**Revised Extended Data Fig. 7**). Take the DEGs in “0 d vs. 7 d” comparison from cerebellum as an example (**Revised Extended Data Fig. 7a-e**). We first evaluated quality metrics such as sequencing depth and library size (**Revised Extended Data Fig. 7a**) and marginal distribution of gene mean and dispersion (**Revised Extended Data Fig. 7b**) for samples in “0 d vs. 7 d” comparison. We then simulated the differentially expressed genes for “0 d vs. 7 d” comparison and evaluated the results of simulated data and actual data to obtain false discovery rate (FDR) (**Revised Extended Data Fig. 7e**). The results show that FDR of the actual detection of DEGs was less than 5%

when we used the same threshold (i.e., $|\log_2FC| > 0.75$) for the simulated data (**Revised Extended Data Fig. 7e**). This indicated that large transcriptional changes in the cerebellum in “0 d vs. 7 d” comparison should be attributed to hypoxia effect instead of technical artifact. Similarly, we used the same analysis procedure, and demonstrated that large transcriptional changes in the colon in “7 d vs. 14 d” comparison (FDR < 0.05) (**Revised Extended Data Fig. 7f-j**) were not technical artifact. Please see **lines 203-208**.

Lines 203-208:

“We performed power analyses to test the credibility of differential expression analyses in comparison of “0 d vs. 7 d” in cerebellum and “7 d vs. 14 d” in colon. The results showed that FDR of the actual detection of DEGs was less than 0.05 when we used the same thresholds (i.e., $|\log_2FC| > 0.75$) for simulated data and actual data (Extended Data Fig. 7), indicating that a large number of DEGs identified in the active tissues are reliable and not technical artifact.”

Following your suggestion, we also conducted differential expression analysis for each tissue in the comparison of “0 d vs. 7 d, 0 d vs. 14 d, 0 d vs. 21 d and 0 d vs. 8 mon” separately (**Supplementary Table 14**). We found that a great number of DEGs detected in cerebellum in all comparisons and in colon from “0 d vs. 14 d” comparison (**Revised Extended Data Fig. 8**). We then performed power analyses for the DEGs in these comparisons with large transcriptional change (**Revised Extended Data Figs. 9,10**), and the results (FDR values < 0.05) excluded the possibility of technical artifact in our findings. Please see **lines 222-230**.

Lines 222-230:

“To further dissect the magnitude of transcriptional change during hypoxia acclimatization, we also identified DEGs across tissues in high altitude Hu sheep at four time points (7 d, 14 d, 21 d and 8 mon) in comparison with low altitude Hu sheep (0 d) (Supplementary Table 14). Similar to the adjacent time comparison, we detected a large number of DEGs in cerebellum in all comparisons and in colon from “0 d vs. 14 d” comparison (Extended Data Fig. 8). Power

analyses evaluated false discovery rate (FDR) of the identified DEGs in the above tissues and found the FDR values were less than 0.05, which support high credibility and exclude technical artifacts in these DEG detection (Extended Data Figs. 9 and 10)."

3) The combination of FST and DEG and DCG genes is interesting, but very ad hoc and not well justified (beyond the urge to identify few genes ...). In addition, it is not clear whether the observed overlap is significant. Permutations of DEG or FST genes could help here

[3.3] Good comment! We would like to interpret the results in **Fig. 4a-c** to make it clear. For each tissue, we first intersected F_{ST} genes with inter-breed DEGs and DCGs separately. We examined the distribution of two categories of intersected genes (i.e., F_{ST} genes in DEGs and F_{ST} genes in DCGs) across tissues, and identified 52 multi-tissue F_{ST} genes in DEGs and 179 multi-tissue F_{ST} genes in DCGs, including 13 common genes which were showed in **Fig. 4c**. Thus, the overlapped genes shown in **Fig. 4c** are quite few (i.e., 13 common genes).

Following your comments, we have conducted permutation test with in-house script. The detailed process was as follows: simulated datasets (i.e., dataset A' and dataset B') were generated randomly as the same size of actual dataset A and dataset B. After 1,000 times shuffle, we compared the number of overlaps between actual dataset A and dataset B with the distribution of overlap statistics between the simulated dataset A' and dataset B', and calculated the statistical significance of P -values (i.e., the probability that higher number of overlaps would be observed by chance).

Take the scenario showed in **Fig. 4c** as an example. We detected a total of 52 multi-tissues F_{ST} genes in DEGs and 179 multi-tissue F_{ST} genes in DCGs separately, and identified 13 common genes. We conducted the above described permutation test to evaluate whether the observed 13 overlapped genes are significant, and obtained

a significant value of $P < 0.001$. Please see **lines 317-318** (“*The 13 common genes are significantly greater than expected by chance (permutation test, $P < 0.001$)*”).

We also conducted permutation test in the following scenarios: 1) the significance of overlaps between F_{ST} genes and all DEGs or all DCGs from 19 tissues, respectively; 2) the significance of overlaps between F_{ST} genes and DEGs or DCGs for each tissue separately. In scenario 1, the results of permutation test showed that overlaps between F_{ST} genes and all DEGs and overlaps between F_{ST} genes and all DCGs were significant ($P < 0.001$) (**Revised Extended Data Fig. 15**). In scenario 2, overlaps between F_{ST} genes and DEGs in each of 14 tissues and overlaps between F_{ST} genes and DCGs in each of 12 tissues were significant ($P < 0.05$) (**Supplementary Table 23**).

I have compiled a non-comprehensive number of small issues that I think could be improved

1) I am confused by the term “plain-to-plateau transplantation” shouldn’t it be rather “plain-to-plateau translocation”? I am not a native speaker, but I think that transplantation should be replaced by translocation throughout the manuscript.

[3.4] Corrected.

2) I think the headings of the subsections should be more informative and consistent

[3.5] Good comments! We have changed the headings of the subsections which can represent the important information in each section. For example, we changed “*Time-series expression changes*” to “*Time-series expression changes reveal high sensitivity of cerebellum in response to hypoxia*”, please **lines 274-275**.

3) I think it is more helpful to report average values with confidence intervals in the text when reporting e.g. SpO2 differences rather than two digit precise very low p-values

[3.6] We have added average values with confidence intervals throughout the manuscript (e.g., **lines 128-130**: “*0 d vs. 7 d, 0 d: 97.3, 95% confidence interval: 96.68 -*

97.92; 7 d: 82.5, 95% confidence interval: 81.87 - 83.15; Wilcoxon rank sum test, $P = 7.50 \times 10^{-13}$).

In addition, we summarized basic statistics (e.g., mean and standard deviation) for phenotypic parameters in **Supplementary Table 11**, please see the details.

4) L. 168: “significantly correlated (FDR < 0.05) with the bio-indicators” should be “significantly correlated (FDR < 0.05) with one or more bio-indicator”

[3.7] Revised.

5) Line 158ff “..only observed a typical pattern of gene expression in the abomasum, which showed an obvious separation of samples before and after 14 d” what is a typical pattern? This is unprecise/unclear.

[3.8] To make the meaning of this pattern clearer, we have replaced “typical” with “particular” to describe the specific feature of clustering result in abomasum. Please see **line 165** (“*We only observed a particular pattern of gene expression in the abomasum*”).

6) At the beginning of the section “Regulation of gene expression by TAD-constrained cis-regulatory elements” it should be made clearer that the authors used previous Hi-C data to assign TADs and from which tissue the TADs were constructed.

[3.9] We have added the details of the Hi-C data, please see **lines 379-381** (“*We obtained 3,032 TADs using a previously published Hi-C data which was generated from the blood of Tibetan sheep²⁸*”).

Reviewers’ comments

Reviewer #4 (Remarks to the Author):

This work examined multi-omics changes of sheep in response to high-altitude stress. The experiments are complete and all analyses are good. Data and cost are surprising. I had thought to use cattle to do the similar analyses and did not do because of the lack of the enough founding. Overall, I am satisfied with experiments

designed and the data collected. There are several questions that need to be addressed.

1. Which breed or group is sister or closely related to Tibet Sheep? Is it Hu Sheep? If not, how is the phylogenetic relationship will affect the identified genes during the low-term evolution? I suggest to use alternative sheep breed to examine such a divergence.

[4.1] Good comments! We would like to interpret how we obtained the identified genes during the low-term evolution. We calculated genome-wide pairwise F_{ST} values between Tibetan sheep and low altitude sheep (including five typical low altitude breeds: Hu sheep, Hu; Wadi sheep, WDS; large tailed Han sheep, HDW; small tailed Han sheep, SXW and Sishui fur sheep, SSS) from our previous study (**Supplementary Table 9**) using a sliding window approach. Based on the SNPs in the regions with the top 5% F_{ST} distribution, we annotated and obtained the identified genes. We want to explain that in the whole-genome selective sweep tests, we used five sheep breeds, rather than only Hu sheep, to represent low altitude breed to compare with Tibetan sheep to obtain the identified genes.

Moreover, to examine the phylogenetic relationship between Tibetan sheep and the five typical low altitude sheep breeds described above (**Supplementary Table 39**), we calculated the identity-by-state (IBS) genetic distances between the individuals of these breeds and visualized the IBS distances by neighbor-joining (NJ) tree. We found two genetic clusters in the NJ tree which are consistent with the geographical distribution of the analyzed breeds from high altitude and low altitude (**Revised Extended Data Fig. 25**). We further computed the F_{ST} distance between Tibetan sheep and each of aforementioned low altitude sheep breeds. The results suggested that Hu sheep (F_{ST} distance = 0.025) and Wadi (F_{ST} distance = 0.025) sheep are most closely related to Tibetan sheep (**Supplementary Table 38**). Therefore, we considered that the low altitude sheep breeds used in the whole-genome selective sweep tests

are appropriate and could not affect the identified genes during the low-term evolution. Please see **lines 583-585**.

Lines 583-585:

“Also, Hu sheep is one of the low altitude sheep breeds which exhibit the closest genetic distance with Tibetan sheep (Extended Data Fig. 25 and Supplementary Table 38)”

2. According to my knowledge, some Hu sheep also occur in the high-altitude region in Sichuan. How do you determine that the Hu sheep materials you used have not been experienced the high-altitude pre-adaptation? Is it possible to give a phylogenetic relationship of the used Hu sheep and those in the other regions?

[4.2] Thanks for your comments. Our Hu sheep were purchased from Huzhou (Zhejiang Province, China), where Hu sheep have been originally distributed, then transported and housed at Wanghu Livestock Farm in Neijiang City (Sichuan Province, China) with an altitude of ~350 m.a.s.l. We then translocated 40 of these Hu sheep to the Tibetan Sheep Breeding Farm of Sichuan Province (Aba Tibetan and Qiang Autonomous Prefecture, Sichuan Province, China) with an altitude of ~3,500 m.a.s.l. Therefore, Hu sheep used in our study have no pre-adaptation to high altitude.

To examine the phylogenetic relationship between Hu sheep and those in other regions, we integrated whole genome sequencing (WGS) dataset newly generated in this round of revision for 49 Hu sheep ewes used in this study (accession No: PRJNA1000743) with 77 WGS datasets from our previous work (**Supplementary Table 39**). Similarly, we calculated IBS between the individuals of sheep breeds and the genetic relationships among individuals were visualized by the NJ tree. We found that our Hu sheep were well clustered with pure Hu sheep from low-altitudes in previous studies (**Revised Extended Data Fig. 24**), supporting that Hu sheep used in this study were from low-altitude plain. Please see **lines 580-583**.

Lines 580-583:

“Hu sheep and Tibetan sheep used in this study are pure breeds that couldn’t affect the whole-genome selection test. From the phylogenetic tree analysis, our Hu sheep were clustered with the pure individuals from low altitude and should have no pre-adaptation at high altitude (Extended Data Fig. 24).”

3. Have Tibet sheep experienced introgression and obtained the related genes from other sheep breeds that had acclimated to the Tibet high-altitude regions earlier? How did you exclude this possibility? If not, how will this will affect your conclusion?

[4.3] To the best of our knowledge, Tibetan sheep is the only sheep breed in China which have well adapted to high-altitude Tibet region. Nevertheless, we integrated 145 WGS datasets of 16 Chinese native sheep breeds from our previous study (**Supplementary Table 39**) with newly generated WGS datasets of 49 Hu sheep ewes to perform introgression analysis. We calculated D statistics for the four-taxon model using AdmixTools (v7.0.2). In the model, we used Menz sheep from Africa as the reference population (H1), Tibetan sheep as the target population (H2) and each population of Chinese native sheep (TCS, Tengchong sheep; MXS, Minxian black fur sheep; SXW, small tailed Han sheep; WDS, Wadi sheep; HDW, large tailed Han sheep; HU, Hu sheep; SSS, Sishui fur sheep; WZS, Ujimqin sheep; THQ, Taihang fur sheep; CLS, Cele black sheep; LOP, Lop sheep; ALS, Altay sheep; KAZ, Kazakh sheep; BSB, Bashibai sheep) as the donor (H3), and we assumed the outgroup (H4) was fixed for the ancestral alleles (**Revised Extended Data Fig. 26a**). Hu sheep population ($n = 49$) in this study were marked with asterisk (**Supplementary Table 39**). The statistical significance of the D statistics was evaluated using a two-tailed Z-test, and $|Z\text{-score}| > 3$ was set as the threshold of statistical significance.

The results of D statistics clearly showed that Tibetan sheep haven’t experienced introgression from other Chinese native sheep breeds (**Revised Extended Data Fig. 26b**). Therefore, we can exclude the possibility of interbreed introgression in our findings. Please see **lines 585-588**.

Lines 585-588:

“For Tibetan sheep, we didn’t detect any interbreed introgression from 16 Chinese sheep breeds (Extended Data Fig. 26) or interspecific introgression from sympatric wild sheep (e.g., argali)⁷² (Extended Data Fig. 27).”

4. Tibet sheep may also obtain their high-altitude variants from the ancestral polymorphisms before their occurrences in the Tibet. They may have obtained such variants through interspecific introgressions. How do you exclude this possibility?

[4.4] Good comments! Following your comments, we used WGS datasets of 17 Tibetan sheep and 7 wild sheep argali (**Supplementary Table 39**) which have sympatric distribution with Tibetan sheep from our previous studies to perform introgression analysis. Similar with the analysis in the comment 3, *D* statistics was calculated in the form (Menz sheep, Tibetan sheep, Argali, Outgroup) (**Revised Extended Data Fig. 27a**), i.e., Menz sheep as the reference population, Tibetan sheep as the target population, Argali as the donor and outgroup was fixed for the ancestral allele. The result showed no interspecific introgression from argali to Tibetan sheep (**Revised Extended Data Fig. 27b**). Please see **lines 585-588**.

Lines 585-588:

“For Tibetan sheep, we didn’t detect any interbreed introgression from 16 Chinese sheep breeds (Extended Data Fig. 26) or interspecific introgression from sympatric wild sheep (e.g., argali)⁷² (Extended Data Fig. 27).”

5. For high-altitude adaptation, reproductive survivals are more important than others, especially for humans. Female Tibetans with Hif1A variants are more likely producing the alive babies when without any modern care. Is it possible to add these reproductive phenotypes when compared them?

[4.5] Good comment! In fact, we have recorded several reproductive phenotypes in our translocated high altitude Hu sheep during the translocation experiment. We summarized them in the **Table R1**. The survival rate of the lambs is approximately

73.68% in translocated high altitude Hu sheep. Besides, we also estimated the survival rate of Tibetan lamb based on the previous records from Tibetan Sheep Breeding Farm of Sichuan Province. The survival rate of Tibetan lamb ranged from 90 % - 95%. We have added the above contents and relevant references (**references 82-84**) in the part of discussion, please see **lines 667-670**.

Table R1. Reproductive record in translocated high altitude Hu sheep in this study.

No. ewes	Litter size	Survive	Died	Total
4	1	4	0	4
3	2	5	1	6
3	3	5	4	9

References

82. Moore, L. G. Fetal Growth Restriction and Maternal Oxygen Transport during High Altitude Pregnancy. *High Altitude Medicine & Biology* 4: 141 – 156 (2003).
83. Neary, J. M. et al. An investigation into beef calf mortality on five high-altitude ranches that selected sires with low pulmonary arterial pressures for over 20 years. *J. Vet. Diagn. Invest.* 25, 210–218 (2013).
84. Jang, E. A., Longo, L. D. & Goyal, R. Antenatal maternal hypoxia: Criterion for fetal growth restriction in rodents. *Front. Physiol.* 6, 176 (2015).

Lines 667-670:

“Compared with humans⁸² and other lowland animals translocated to high altitudes (e.g., cattle⁸³ and mouse⁸⁴), sheep showed much less fetal growth restriction and neonatal mortality based on both the observation made in our translocation experiment (neonatal mortality: 26.32%)”

REVIEWER COMMENTS

Reviewer #1 (Remarks to the Author):

The authors have addressed all of my comments and those of the other reviewers in a reasonable way. I anticipate that the results of this extremely large, comprehensive study will be of wide value to other researchers and will support greater insight in to the mechanisms of adaptation to hypoxia in the future.

Reviewer #1 (Remarks on code availability):

This does not seem relevant to this study.

Reviewer #2 (Remarks to the Author):

The authors have a significant effort and improved the clarity of the manuscript. This is a very interesting resource for the community, in particular now with the data in the repository in a better format. The authors addressed all my concerns in the revised version

Reviewer #3 (Remarks to the Author):

The authors have adressed my concerns

Reviewer #4 (Remarks to the Author):

The revised version includes a significant amount of additional data and analyses that greatly improve the overall conclusion. While most of my concerns have been addressed, there are still some areas that require further clarification.

1. The sections Introduction and Discussion did not establish a clear logical connection between short-term adaptation and long-term adaptation. This gap in understanding may lead to a misunderstanding of the manuscript as a whole. From my understanding, the differentially expressed genes (DEGs) identified during short-term adaptation should have corresponding allelic single nucleotide polymorphisms (SNPs) or structural variations (SVs) between high-altitude and low-altitude populations during long-term adaptation. The long-term selection process would favor individuals with allelic SNPs and SVs that correspond to DEGs during the short-term adaptation. This point should be explicitly and succinctly stated.

2. Regarding the possibility of interspecific introgression favoring the long-term high-altitude adaptation of sheep, I have reservations about the current simplistic tree construction method. The presence of only a few introgressed genes is unlikely to significantly affect the overall tree topology. In fact, commonly used methods such as Structure analyses, ABBA-BABA analyses, and gene trees of each gene should be employed.

3. Even though the new analyses did not detect introgression in the long-term high- and low-altitude populations, the authors should still discuss the possibility of introgression as it cannot be completely ruled out for two reasons. First, the sample size may not be sufficient. Second, the present analyses only utilized SNPs and did not consider all genetic variations. To support my review of this manuscript, I have explored recent papers and found that introgressions between yak and cattle related to altitude adaptation, as described in Nature Communications, were only revealed based on SVs in a few key genes, such as EPAS1. The authors should address these limitations and potentialities.

4. I also recommend condensing the current Discussion section while incorporating all the aforementioned limitations as suggested earlier. Especially, SVs (including plants, for example, Arabidopsis during my looking for recent progresses) were found to contribute to the long-time high-altitude adaptation of both plants and animals. Allelic SVs in the promoters directly result in differentially expressions of the key genes. All related recent references should be integrated to discuss the caveats and possibilities.

REVIEWER COMMENTS

Reviewer #1 (Remarks to the Author):

The authors have addressed all of my comments and those of the other reviewers in a reasonable way. I anticipate that the results of this extremely large, comprehensive study will be of wide value to other researchers and will support greater insight in to the mechanisms of adaptation to hypoxia in the future.

Reviewer #1 (Remarks on code availability):

This does not seem relevant to this study.

Thank you for your comment! We have modified the description of the computational scripts and codes at GitHub repository "<https://github.com/zeyan-0717/sheep-transcriptome-atlas>", so that the content presented in the "code availability" could be clearly relevant to this study.

Reviewer #2 (Remarks to the Author):

The authors have a significant effort and improved the clarity of the manuscript. This is a very interesting resource for the community, in particular now with the data in the repository in a better format. The authors addressed all my concerns in the revised version

Thanks!

Reviewer #3 (Remarks to the Author):

The authors have addressed my concerns

Thanks!

Reviewer #4 (Remarks to the Author):

The revised version includes a significant amount of additional data and analyses that greatly improve the overall conclusion. While most of my concerns have been addressed, there are still some areas that require further clarification.

1. The sections Introduction and Discussion did not establish a clear logical connection

between short-term adaptation and long-term adaptation. This gap in understanding may lead to a misunderstanding of the manuscript as a whole. From my understanding, the differentially expressed genes (DEGs) identified during short-term adaptation should have corresponding allelic single nucleotide polymorphisms (SNPs) or structural variations (SVs) between high-altitude and low-altitude populations during long-term adaptation. The long-term selection process would favor individuals with allelic SNPs and SVs that correspond to DEGs during the short-term adaptation. This point should be explicitly and succinctly stated.

[4.1] Good comment! Following your suggestion, we have added the point regarding the connection between short-term acclimatization and long-term adaptation in the sections Introduction (**lines 46-51**), Results (**lines 344-350**) and Discussion (**lines 606-613**). To support the description of the point, we cited relevant reference papers (**references 9-11 and 79**) and performed the permutation test ($n = 1,000$) to examine whether the number of overlapped genes between long-term adaptation (i.e., F_{ST} genes) and short-term acclimatization (i.e., DEGs) is significantly higher than that expected at random. We detected 2,548 F_{ST} genes and 13,891 DEGs from 19 tissues, and identified 1,770 overlapped genes which is significantly higher than that expected at random (i.e., 1,234 genes, $P < 0.001$) (**Extended Data Fig. 15**), supporting that long-term selection process would favor individuals with allelic SNPs that correspond to DEGs during the short-term adaptation.

References:

9. Friedrich, J. & Wiener, P. Selection signatures for high-altitude adaptation in ruminants. *Animal Genetics* 51, 157–165 (2020).
10. Liu, X. et al. Evolutionary origin of genomic structural variations in domestic yaks. *Nat. Commun.* 14, 5617 (2023).
11. Chen, X. et al. Comparison between short-term stress and long-term adaptive responses reveal common paths to molecular adaptation. *iScience* 25, 103899 (2022).
79. Price, T. D., Qvarnström, A. & Irwin, D. E. The role of phenotypic plasticity in driving genetic evolution. *Proceedings of the Royal Society of London. Series B: Biological Sciences* 270, 1433–

1440 (2003).

Lines 46-51:

“Both short-term acclimatization and long-term adaptation have a genetic background, thus they are genetically linked to each other⁹. For instance, differentially expressed genes (DEGs) identified during short-term acclimatization may have corresponding SNPs or structural variants (SVs) between high-altitude and low-altitude populations during long-term adaptation¹⁰. Common signaling pathway may be enriched during short-term response and long-term adaptation to hypoxia¹¹.”

Lines 344-350:

“Additionally, we conducted permutation test to examine whether the overlaps between F_{ST} genes and all DEGs within Hu sheep from 19 tissues were significantly higher than that expected at random. The result showed that 1,770 overlapped genes between long-term adaptation (i.e., 2,648 F_{ST} genes) and short-term acclimatization (i.e., 13,891 DEGs) is significantly higher than that expected at random (i.e., 1,234 genes, $P < 0.001$) (Extended Data Fig. 15), implying that long-term selection may favor individuals with SNPs that correspond to DEGs during short-term acclimatization.”

Lines 606-613:

“Moreover, we found that the overlapped genes between long-term adaptation and short-term acclimatization are significantly higher than the random overlaps (Extended Data Fig. 15). Previous studies usually investigated short-term and long-term responses to environmental stress separately, but the interaction between the two processes has been rarely evaluated, e.g., whether short-term transcriptional response may facilitate or constrain long-term genetic adaptation^{11,79}. Our results provided a valuable example that short-term acclimatization could facilitate long-term adaptation to high-altitude hypoxia.”

2. Regarding the possibility of interspecific introgression favoring the long-term high-altitude adaptation of sheep, I have reservations about the current simplistic tree

construction method. The presence of only a few introgressed genes is unlikely to significantly affect the overall tree topology. In fact, commonly used methods such as Structure analyses, ABBA-BABA analyses, and gene trees of each gene should be employed.

[4.2] Thank you for your comment. We would like to interpret that in the last round of revision, we have already conducted ABBA-BABA analyses (i.e., *D* statistics) to detect potential interspecific introgression using WGS datasets of 17 Tibetan sheep and 6 sympatric wild sheep argali (Supplementary Table 39) from our previous studies with four-taxon model (Menz sheep, Tibetan sheep, Argali, Outgroup) (**Revised Extended Data Fig. 29a**). The result showed no interspecific introgression from argali to Tibetan sheep (**Revised Extended Data Fig. 29b**).

Following your comment, we performed Structure analyses and constructed gene trees with the same samples in the ABBA-BABA analyses (Supplementary Table 39). For the Structure analyses, we conducted model-based clustering analyses using sNMF software with the number of *K* from 1 to 20. The optimal *K* (i.e., *K* = 2) was determined by the minimal value of the cross-entropy (**Revised Extended Data Fig. 28a**). When *K* = 2, the samples were separated into two distinct clusters, which correspond to argali and Tibetan sheep (**Revised Extended Data Fig. 28b**).

We also examined the potential interspecific introgression between argali and Tibetan sheep among the top selective region identified between low altitude sheep and Tibetan sheep. We extracted the SNPs located in the genomic regions encompassing the top 5 F_{ST} gene (i.e., 50 kb up and downstream), and then constructed gene tree for each of these genes with SplitsTree. The results showed that samples were split into two separate branches (i.e., Tibetan sheep and argali) in the gene trees (**Revised Extended Data Fig. 30**). Based on the findings from the above analyses, we excluded the possibility of interspecific introgression in our Tibetan sheep. Please see **lines 692-699**.

Lines 692-699:

“For Tibetan sheep, we didn’t detect any interbreed introgression from 16 Chinese sheep breeds (Extended Data Fig. 27) or interspecific introgression from sympatric wild sheep (e.g., argali)⁹³ based on the Structure (Extended Data Fig. 28), ABBA-BABA (Extended Data Fig. 29) and gene tree analyses (Extended Data Fig. 30). Thus, we excluded the potential impact of phylogenetic relationship or genetic introgression on the whole-genome selection test between high altitude Tibetan sheep and low altitude sheep (e.g., Hu sheep) in our study.”

3. Even though the new analyses did not detect introgression in the long-term high- and low-altitude populations, the authors should still discuss the possibility of introgression as it cannot be completely ruled out for two reasons. First, the sample size may not be sufficient. Second, the present analyses only utilized SNPs and did not consider all genetic variations. To support my review of this manuscript, I have explored recent papers and found that introgressions between yak and cattle related to altitude adaptation, as described in *Nature Communications*, were only revealed based on SVs in a few key genes, such as EPAS1. The authors should address these limitations and potentialities.

[4.3] Thank you for your comment! We have added discussion for the limitations and potentialities regarding the possibility of introgression in our study based on your suggestion. The relevant reference paper in *Nature Communications* (**reference 10**) has been also cited. Please see **lines 699-706** in the revised version.

Reference:

10. Liu, X. et al. Evolutionary origin of genomic structural variations in domestic yaks. *Nat. Commun.* 14, 5617 (2023).

Lines 699-706:

“Nevertheless, we cannot completely rule out the possibility of introgression and its effect on high altitude adaptation of Tibetan sheep. First, although introgression was not detected in Tibetan sheep used in this study, such introgression may be identified in future studies when

more individuals of Tibetan sheep are included and analyzed. Second, our introgression analyses only utilized SNPs and did not consider other types of genetic variations (e.g., SVs) in some specific functional genes. A recent study found that the introgressed segments between yak and cattle related to altitude adaptation were only revealed based on SVs in a few key genes (e.g., EPAS1)¹⁰.”

4. I also recommend condensing the current Discussion section while incorporating all the aforementioned limitations as suggested earlier. Especially, SVs (including plants, for example, Arabidopsis during my looking for recent progresses) were found to contribute to the long-time high-altitude adaptation of both plants and animals. Allelic SVs in the promoters directly result in differentially expressions of the key genes. All related recent references should be integrated to discuss the caveats and possibilities.

[4.4] Following your comment, we condensed the current Discussion section by removing the discussions for less important results, and consequently around 240 words were deleted. We also incorporated discussions about the limitations and possibilities suggested in this comment (e.g., SVs could contribute to the long-time high-altitude adaptation of both plants and animals, and allelic SVs in the promoters directly result in changes in gene expression) and aforementioned in comment 3 into the Discussion section, and cited recent references (**references 10 and 94**) related to the discussions. Please see **lines 706-717** in the revised version.

References:

10. Liu, X. et al. Evolutionary origin of genomic structural variations in domestic yaks. *Nat. Commun.* 14, 5617 (2023)
94. Kang, M. et al. The pan-genome and local adaptation of Arabidopsis thaliana. *Nat. Commun.* 14, 6259 (2023).

Lines 706-717:

“SVs could contribute to the high altitude adaptation of both animals (e.g., yak)¹⁰ and plants (e.g., Arabidopsis thaliana)⁹⁴. For instance, an insertion in the promoter region of the HPCA1

gene enhances gene expression and promotes the adaptation of Arabidopsis thaliana to alpine environments⁹⁴. Despite the importance of SVs, it is unfortunate that we don't have long-read sequencing genomic data and a large panel of samples to yield precise and comprehensive SVs for corresponding analysis in this study. Also, the focus of the present study is to reveal transcriptional regulation in response to high-altitude hypoxia, therefore genomic analysis on SVs is beyond our main topic. Future studies focusing on SVs may discover whole-genome SVs characteristics and the impact of SVs on introgression, gene expression and high altitude adaptation in sheep."

REVIEWERS' COMMENTS

Reviewer #4 (Remarks to the Author):

I have no further concerns and the present version is sufficient for being published.